# Large dynamics of a phase separating arginine-glycine-rich domain revealed via nuclear and electron spins

Giuseppe Sicoli [1,5], Daniel Sieme [2,5], Kerstin Overkamp[2], Mahdi Khalil [1], Robin Backer [3], Christian Griesinger[2], Dieter Willbold[3,4] & Nasrollah Rezaei-Ghaleh [3,4] ✉

Liquid-liquid phase separation is the key process underlying formation of membrane-less compartments in cells. A highly dynamic cellular body with rapid component exchange is Cajal body (CB), which supports the extensive compositional dynamics of the RNA splicing machinery, spliceosome. Here, we select an arginine-glycine (RG)-rich segment of coilin, the major component of CB, establish its RNA-induced phase separation, and through combined use of nuclear magnetic resonance (NMR) and electron paramagnetic resonance (EPR) probes, interrogate its dynamics within the crowded interior of formed droplets. Taking advantage of glycine-based singlet-states, we show that glycines retain a large level of sub-nanoseconds dynamics inside the coilin droplets. Furthermore, the continuous-wave (CW) and electron-electron dipolar (PELDOR) and electron-nucleus hyperfine coupling EPR data (HYSCORE) support the RNA-induced formation of dynamic coilin droplets with high coilin peptide concentrations. The combined NMR and EPR data reveal the high dynamics of the RG-rich coilin within droplets and suggest its potential role in the large dynamics of CBs.

Cellular activities frequently occur within membrane-less compartments, which are formed through a physicochemical process termed liquid–liquid phase separation (LLPS) in cells[1]. Several types of cellular bodies are formed via the LLPS process, e.g. processing or P-bodies, stress and germ granules in the cytoplasm and nucleoli, Cajal (coiled) bodies (CB) and nuclear speckles (NS) in the nucleus[2]. A remarkable feature of the non-membranous organelles is that they are highly dynamic and can be formed or dissolved rapidly in response to environmental changes and cellular cues[2]. In addition, they exchange their components with the surrounding environment much faster than the membranous compartments[3].

The key physicochemical principle underlying biomolecular LLPS is multivalency, which enables generating an interaction network as a separate phase. Multivalent interactions in proteins can occur between modular proteins containing repetitive folded domains and intrinsically disordered low-complexity domains (LCDs), or alternatively, between intrinsically disordered proteins (IDPs) capable of multiple weak interactions. Frequently, the LLPS of proteins occurs as complex coacervation, often promoted by a poly-ion compound such as RNA[4]. A ubiquitous evolutionary conserved RNA interaction motif in ribonucleoproteins is the Arg/Gly-rich (RG-rich) LCD[5–7].

The internal fluidity of biological condensates is encoded in the amino acid and nucleotide sequence of the involved proteins and

[1]CNRS UMR 8516, University of Lille, LASIRE, C4 Building, Avenue Paul Langevin, F–59655 Villeneuve d'Ascq, France. [2]Department of NMR-based Structural Biology, Max Planck Institute for Multidisciplinary Sciences, Am Faßberg 11, D-37077 Göttingen, Germany. [3]Heinrich Heine University (HHU) Düsseldorf, Faculty of Mathematics and Natural Sciences, Institute of Physical Biology, Universitätsstrasse 1, D-40225 Düsseldorf, Germany. [4]Institute of Biological Information Processing, IBI-7: Structural Biochemistry, Forschungszentrum Jülich, Wilhelm-Johnen-Straße, D-52428 Jülich, Germany. [5]These authors contributed equally: Giuseppe Sicoli, Daniel Sieme. ✉e-mail: Nasrollah.Rezaie.Ghaleh@hhu.de

RNAs and their concentrations, among other factors[8,9]. Residues such as glycine and lysine seem to act as spacers and enhance fluidity, while residues such as glutamine, serine and arginine act more like stickers and thereby promote hardening of the separated phases[10,11]. Material properties such as viscosity, interfacial tension and shear relaxation are controlled through short-range interactions dictated by the sequence of polypeptide and RNA chains and in turn regulate other properties such as internal diffusion, droplet fusion and rates of turn-over (formation-dissolution) and component exchange[12–14]. Importantly, the mis-regulation of cellular condensates and their properties is associated to diseases such as cancers and neurodegenerative and infectious diseases[15].

NMR spectroscopy is a powerful tool for studying the structural and dynamical properties of biomolecules in different phases at atomic resolution. In particular, NMR is sensitive to reorientational dynamics of proteins over a broad range of timescales from pico-to-milliseconds timescales[16,17]. Some phase-separating IDPs are shown through NMR to retain their disordered structure within the liquid-like phase separated assemblies[18–21], while others undergo a partial structural transition[22]. On the other hand, the $^{15}N$ NMR relaxation rates have suggested that global rotational mobility of proteins is considerably restricted in the condensed phase, whereas local protein mobility remains relatively intact[18]. Notably, the imposed restriction in the rotational mobility of proteins is less drastic than that of translational mobility, which may be hindered by 2–3 orders of magnitude[10,18,20]. Recently, an exchange-based NMR method has been employed to monitor and quantify component exchange between different phases[23].

Another technique providing high-resolution structural and dynamical information on biomolecular systems is Electron Paramagnetic Resonance (EPR). EPR spectroscopy has been extensively used to study the dynamics associated with membranes[24,25], liquid crystals[26], micellar solutions[27], proteins[28], and LLPS in polymers and proteins[29–33]. More recently, pulsed-EPR techniques have been exploited within the context of LLPS transition[33,34], confirming the promising applicability of long-range inter-spin distances to monitor formation of droplets.

Prototypical examples of highly dynamic cellular bodies with rapid turn-over and fast component exchange are CBs and NSs, two major sites for the storage, modification, assembly and recycling of RNA-splicing factors[35,36]. Little, however, is known about the molecular factors underlying their remarkably large dynamics. A major component of CBs is coilin, a protein which has no known function other than being the scaffold of CBs[37]. Coilin (UniProt code: P38432) is a 576 residues-long protein, containing a C-terminal folded domain called Tudor domain and an N-terminal region with a propensity for coilin-coilin self-interaction. Coilin contains regions enriched with Lys or Ser and several repeats of RGG/RG motifs. Here, we investigate the hypothesis that the unstructured part of coilin protein contributes to the LLPS-driven formation of CBs and supports their high internal fluidity. Using an RG-rich segment of the coilin protein, supposed to act as the main region involved in the phase separation of coilin, we develop a combined NMR and EPR approach and investigate the pico-to-nanoseconds coilin peptide dynamics within the crowded interior of the formed condensates. In particular, we employ glycine-based NMR singlet states as dynamical probes and demonstrate that several glycine residues of the selected coilin peptide almost entirely preserve their dynamics on hundreds of picoseconds timescale inside the formed droplets. Besides, we provide pulsed EPR data, including for the first time the inter-molecular electron-nucleus hyperfine coupling (HYSCORE) data, supporting the RNA-induced formation of dynamic coilin droplets. Our combined NMR and EPR data reveal the propensity of coilin peptide for RNA-induced phase separation and the large level of reorientational peptide dynamics inside the formed droplets and

highlight its potential to contribute to the internal fluidity of CBs inside cells.

## Results

### Sequence-based selection of a phase-separating fragment of coilin protein

First, we predicted the propensity of coilin for RNA-mediated granule formation[38]. As shown in Fig. 1a, a 36-residue RG-rich segment of coilin protein encompassing residues 389-424 was predicted to possess a high propensity of RNA-mediated granule formation. Accordingly, we selected this fragment of coilin for this study. The selected coilin peptide (clp) contains six RG repeats, four of them tandem repeats, and is predicted to have an isoelectric pH (pI) of 12.1 and a net charge of around +7 at neutral pH.

As expected for the large net charge of clp and the resultant repulsive electrostatic interactions, no phase separation was observed for the peptide solutions alone even at concentrations as high as 0.5 mM. Furthermore, a salt (NaCl) concentration rise to 500 mM combined with sodium phosphate buffer concentration of 50 mM was not sufficient to screen-out the repulsive electrostatic interactions between clp molecules and drive them towards phase separation. Upon addition of 1 mg/mL polyuridylic acid (polyU RNA, ~3.3 mM monomer concentration), however, a rapid induction of phase separation was observed at 0.4 mM clp concentration (hence poly-U:peptide ratio of ~ 8:1), 150 mM NaCl and 50 mM sodium phosphate (pH 6.5). Considering the −1 charge of each unit of polyU and +7 (net) charge of each clp molecule, the ratio of negative charges of polyU to positive charges of clp ($x$) was around 1.1, which, as shown in ref. 12, is well inside the re-entrant liquid condensation regime for RG-rich peptides and close to the isoelectric point, $x_0$, of 1, at which phase separation is expected to be maximal. The occurrence of phase separation was evident by inspection as the formation of a turbid solution (Fig. 1b), and formation of a large number of spherical droplets with radii between 0.46–1.71 μm, as revealed by laser scanning microscopy (Fig. 1c). Furthermore, the use of a nucleic acid-binding fluorescent dye (SYBR Gold) confirmed the presence of RNA in the formed droplets (Fig. 1d). In addition, the combined use of fluorescently-labeled clp (with Atto647) and polyU-binding fluorescent dye confirmed the co-localization of clp and polyU in the formed droplets (Supplementary Fig. 1). The calculated fluorescence intensity ratio between inside and outside of droplets was 204 ± 49 for Atto647-labeled clp and 55 ± 18 for SYBR Gold-bound polyU (Supplementary Fig. 1), indicating that clp and polyU were almost entirely recruited to the formed droplets. The formed droplets exhibited a large level of Brownian mobility in solution and a high rate of droplet-droplet encounters, which interestingly led to the formation of relatively persistent droplet "twins" (Fig. 1e, Supplementary Movie 1). Despite the relatively long lifetime of droplet twins, however, they were often not capable of fusing with each other (Supplementary Movie 2). This behavior is probably due to RNA:peptide ratio being slightly greater than the isoelectric point at which charge inversion occurs. Consequently, the polyU-clp droplets are expected to possess a small net negative charge. In such conditions, the combination of long-range repulsive and short-range attractive electrostatic interactions leads to the kinetic stability of a colloid-like cluster phase where particles aggregate but do not coalesce[12]. Accordingly, when the polyU-clp droplets were re-examined after 3 days, they were found in pearls-string-like arrangements without coalescence (Fig. 1f). Control experiments with only RNA (or clp) did not show any droplet (Fig. 1g).

### RNA-induced droplet formation of coilin peptide monitored through NMR

Next, we utilized NMR to monitor clp before and after polyU-induced droplet formation. The 1D $^1H$ and 2D $^1H,^1H$ TOCSY and NOESY and natural abundance $^{15}N,^1H$ and $^{13}C,^1H$ HSQC spectra (Supplementary

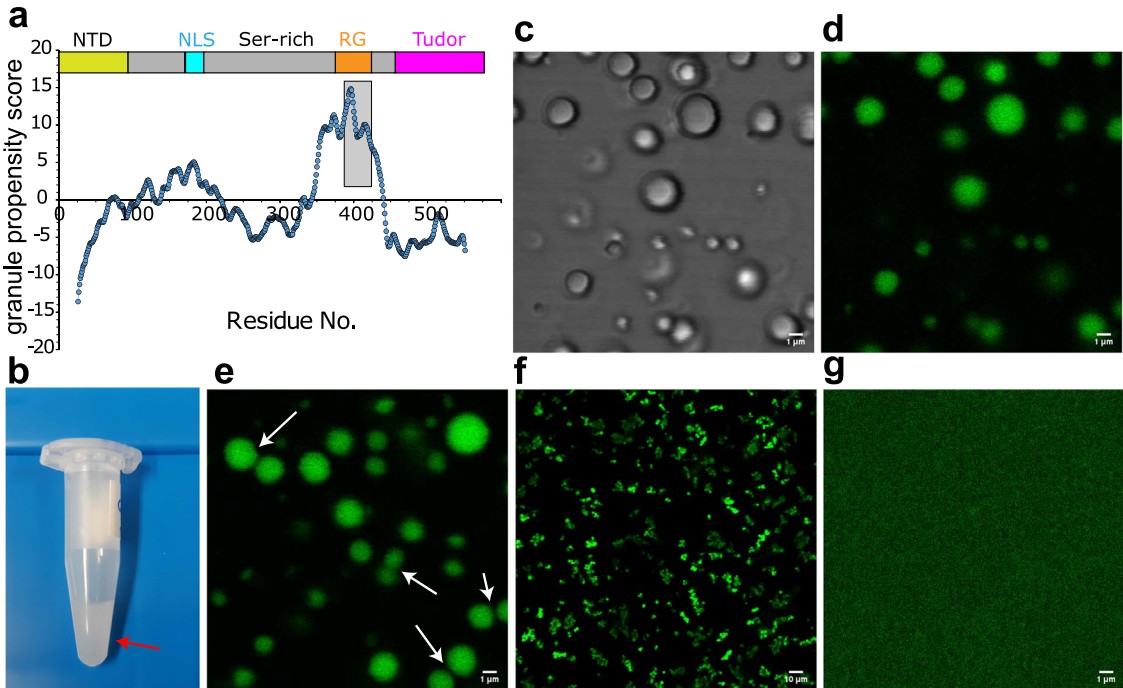

**Fig. 1 | Liquid–liquid phase separation (LLPS) of coilin. a** Sequence-based prediction of the propensity of coilin for RNA-mediated granule formation and LLPS. The shaded area shows the RG-rich region (residues 389-424) with high LLPS propensity. **b-d** The coilin peptide (clp) encompassing residues 389-424 undergoes LLPS after addition of polyU RNA, as revealed by visible turbidity in (**b**) and presence of droplets in light (**c**) and fluorescence microscopy (**d**). The fluorescent nucleic acid-binding dye (SYBR Gold) was used for tracking polyU RNA. In (**e**), the arrows show a few droplet pairs staying close to each other for a long time without being fused. **f** Fluorescence microscopy images of polyU-induced clp droplets after 3 days, showing droplets in pearls-string-like arrangements without fusion. **g** The control sample with polyU without clp did not show any droplet formation.

Figs. 2–5) of clp were measured and utilized for NMR peak assignments. Except for the segment G416-R417-G418-R419, the unambiguous assignment was obtained for all backbone resonances of clp (Fig. 2a). The NMR spectra showed characteristic features of intrinsically disordered peptides (e.g. narrow amide and methyl proton chemical shift dispersion and lack of long-range NOE peaks), and the correlation peaks belonging to the glycine and arginine residues were observed in the characteristic regions of the 2D correlation spectra (Supplementary Figs. 3–5). The comparison of backbone chemical shifts of clp with random coil chemical shifts suggested that clp is predominantly an intrinsically disordered peptide, with weak secondary structure propensities (Fig. 2b).

Upon addition of 1 mg/mL polyU RNA, the 1D $^1$H spectrum showed significant reduction in the intensity of clp signals, especially in the downfield amide region, along with a number of new peaks belonging to polyU (Fig. 3a). The signal intensity decrease reflects the polyU-induced droplet formation by clp, where clp interaction with polyU in the crowded viscous interior of droplets restricts the rotational mobility of clp and consequently causes a severe broadening of its signals beyond NMR detection limit. The high ratio of polyU:clp used (8:1, see above) and the near complete loss of clp's NMR signals observed (Fig. 3a) suggest that the clp molecules are almost entirely recruited to the formed droplets.

To further investigate the recruitment of clp molecules to polyU-induced droplets, we used a mixture of diamagnetic and paramagnetic (2,2,6,6-tetramethyl-N-oxyl-4-amino-4-carboxylic acid- or TOAC-labeled) clp at a dia:para ratio of 9:1 and measured NMR spectra before and after addition of polyU. As shown in Fig. 3b, addition of polyU led to an almost complete loss of clp *and* polyU signals (compare polyU signals in Fig. 3a, b). The almost complete loss of clp and polyU signals after phase separation (Fig. 3b) indicates nearly complete recruitment of clp and polyU molecules to droplets, which brings the paramagnetic TOAC-labeled clp molecules close to diamagnetic clp and polyU

molecules and consequently causes a paramagnetic relaxation enhancement (PRE)-induced broadening of most of their NMR signals. A similar behavior has been recently reported for the RNA-induced phase separation of an RGG-rich peptide, for which Raman microscopy data demonstrated the full peptide recruitment into formed droplets[39]. In 2D $^1$H,$^1$H TOCSY spectrum of the mixture of dia/paramagnetic clp and polyU, the HN-HA correlation peaks from the C-terminal residues V423 and S424, and to a lower extent, several other residues especially close to the C-terminus of peptide, partially survived, indicating their relatively high residual reorientational mobility inside the formed droplets (Fig. 3c). The alternative possibility that these signals were originated from residual clp within the dilute phase can be excluded considering that the (partial) preservation of NMR signals was not uniform along the clp sequence.

## Large sub-nanoseconds dynamics of glycines inside coilin peptide droplets

Subsequently, to more closely monitor the reorientational dynamics of clp, we exploited its large glycine content spread over the sequence (10 residues, Fig. 4a) and employed the recently developed glycine-based singlet-filtered approach[40]. Singlet states are effective spin-0 states formed in homonuclear spin-1/2 pairs, which can be detected indirectly[41]. This approach enables selective detection of the glycine HA signals in non-labeled biomolecular mixtures depending on the ratio between the chemical shift difference of their two HAs and the geminal HA-HA coupling, i.e. $\varepsilon = \Delta v/J$, both in Hz[40]. Due to their exchange symmetry property, singlet-states are protected from several relaxation mechanisms and could therefore possess lifetimes longer than the magnetization relaxation time $T_1$, hence called long-lived states[42]. The glycine-based singlet-state relaxation time ($T_s$) is however sensitive to the dynamics of glycine residues at pico-to-nanoseconds timescale and it has recently been demonstrated that the $T_s/T_1$ ratio is a sensitive probe of peptide dynamics in the

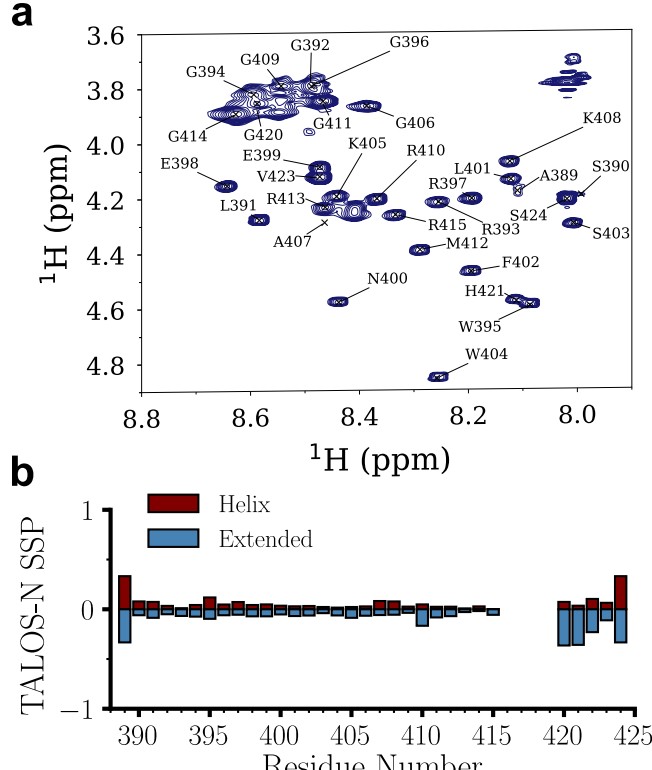

**a**

**b**

**Fig. 2 | Resonance assignment of the coilin peptide (clp). a** Assigned Hᵉ-Hᴺ region from the 2D ¹H–¹H TOCSY spectrum. The intra-residue peaks in the TOCSY were used for amino-acid residue-typing while the additional peaks apparent in the 2D ¹H-¹H NOESY spectrum provided inter-residue information (see also Supplementary Figs. 3–5). **b** Secondary structure propensities determined from the assigned chemical shifts via TALOS-N (top). Propensity for helix (red) and extended (blue) structures are shown respectively on positive and negative sides of Y-axis. Source data are provided as a Source Data file.

"intermediate" motional regime where rotational correlation time ($\tau_c$) is around $1/\omega_H$ (at proton Larmor frequency of 600 MHz, it is ~265 ps) and the $T_s$ and $T_1$ relaxation times move in opposite directions in response to dynamical alterations in $\tau_c$[43–45].

On this background, we first showed that the glycines of clp could be detected and classified in (at least) four groups depending on their ε (Fig. 4b and Supplementary Table 1). For all detected glycine residues, the HA pairs were within intermediate-coupling regime with ε ranging from 0.7 to 2.7, and the search for glycines with HA pairs in strong (ε < 0.2) or weak (ε > 5) coupling regimes did not provide any additional signal. The strongest signals were observed for groups I and II with ε of 0.7 and 1.3, respectively. Through comparison with partially assigned HA-HN correlation peaks of glycines (Fig. 2a) and the singlet-filtered sf-TOCSY experiments (Supplementary Fig. 6), a tentative assignment was obtained for groups I (G414, G420), II (G392, G394, G396), III (411) and IV (406).

Next, we measured the singlet-state and longitudinal relaxation times $T_s$ and $T_1$ for the four groups of glycine residues I-IV at 298 K (Fig. 4c and Table 1). The obtained $T_s$ and $T_1$ represent the collective relaxation behavior of glycines of each group at this temperature. As shown in Table 1, the $T_1$ values of the four glycine groups were highly similar. On the other hand, the $T_s$ values were considerably different, ranging from 1.79 ± 0.03 and 1.82 ± 0.23 s for groups I and III to 2.08 ± 0.08 and 2.23 ± 0.51 s for groups II and IV. The $T_s/T_1$ ratio was 1.84 ± 0.08, 2.14 ± 0.09, 1.95 ± 0.26 and 2.40 ± 0.55, respectively for groups I–IV, all being greater than 1 indicating the long-lived character of the generated spin states. Interestingly, after decreasing temperature to 278 K, all four glycine groups exhibited a significant decrease in

$T_s$ and increase in $T_1$, leading to ca. 2.5–3 fold decrease in $T_s/T_1$ ratio (Table 1). The opposite response of $T_s$ and $T_1$ to temperature drop suggested that the reorientational dynamics of glycines underlying $T_s$ and $T_1$ relaxation occurred largely within the "intermediate" regime with $\tau_c$ close to but longer than $1/\omega_H$ of ca. 256 ps.

Upon addition of polyU RNA to clp and its resultant droplet formation, the glycine singlet-states in groups I and II were still accessible, thanks to efficient filtering out of the other signals from the peptide and particularly from polyU (Fig. 4d). On the other hand, the glycine singlet-states in groups III and IV were too weak to be detected in the formed condensed phase. The signals of glycine groups I and II were originated from within droplets, because, as stated above, nearly all clp molecules had been recruited to droplets. Strikingly, however, the measured $T_1$, $T_s$ and $T_s/T_1$ values of glycine groups I (G414, G420) and II (G392, G394 and G396) in the polyU-induced clp droplets were within the experimental range of error identical with those obtained before addition of polyU (Table 1). Only the $T_1$ of glycine group II was marginally higher in presence of polyU, possibly reflecting a slight rigidification of this N-terminal region. Overall, the singlet-state and $T_1$ relaxation data indicates that glycine residues retain a large level of their local sub-nanoseconds reorientational dynamics in the crowded interior of clp droplets. It is noteworthy that the assigned glycine residues belonging to groups I and II (G392, G394, G396, G414, G420) are spread over the clp sequence, suggesting that glycine residues along clp sequence may undergo high local reorientational dynamics inside droplets.

### Diminished dynamics of tryptophan sidechains inside coilin peptide droplets

To gain further insight on local dynamics of clp in dilute and polyU-induced condensed phases, we studied the ¹H $T_1$ relaxation of tryptophan sidechains in clp. The clp has two tryptophans, W395 and W404, and the nitrogen-attached protons of the indole ring of these two residues are well resolved at 10.13 and 10.21 ppm (Supplementary Fig. 7). The ¹H $T_1$ relaxation time of W395 and W404 obtained through saturation-recovery measurements at 298 K were 0.28 ± 0.01 s and 0.24 ± 0.01 s, respectively (Fig. 4e and Supplementary Fig. 7). The temperature-dependence of ¹H $T_1$ suggested that reorientational dynamics of tryptophan sidechains occur at $\tau_c$ longer than $1/\omega_H$ of ca. 256 ps (Supplementary Fig. 7). Upon addition of polyU and the consequent droplet formation by clp, the ¹H $T_1$ of W395 and W404 increased respectively to 0.44 ± 0.09 s and 0.41 ± 0.11 s (Fig. 4e and Supplementary Fig. 7), indicating that the reorientational dynamics of the two tryptophan sidechains in the N-terminal half of clp were lowered inside the condensed phase.

### Tuning of dynamics for spin-labeled coilin peptide upon formation of droplets: EPR data

To investigate the LLPS of coilin by EPR, nitroxide spin labels TOAC, MTSSL and PROXYL have been selected as paramagnetic probes; they were inserted either at the terminal residue (MTSSL) or inserted between the first two N-terminal residues (Ala389 and Ser390) of clp (TOAC, PROXYL) (Fig. 5a, Supplementary Fig. 8). The CW EPR spectra were measured at room temperature before and after polyU-mediated LLPS, as in the NMR experiments. The nitroxide CW spectrum is particularly sensitive to the local environment and gives information on backbone and side chain fluctuation at the labeling site[46]. The initial CW EPR spectra of clp and clp/polyU recorded at room temperature exhibited different spectral features for the TOAC spin probe (Supplementary Fig. 8); however, the signals obtained with the paramagnetic probe inserted into the peptide chain without additional alkyl linkers is affected by very low signal-to-noise ratio (S/N), especially for the sample containing polyU. Although the obtained S/N is not optimal, it is clear that rotational correlation times and spectral widths are different when coilin undergoes LLPS. By using labels with

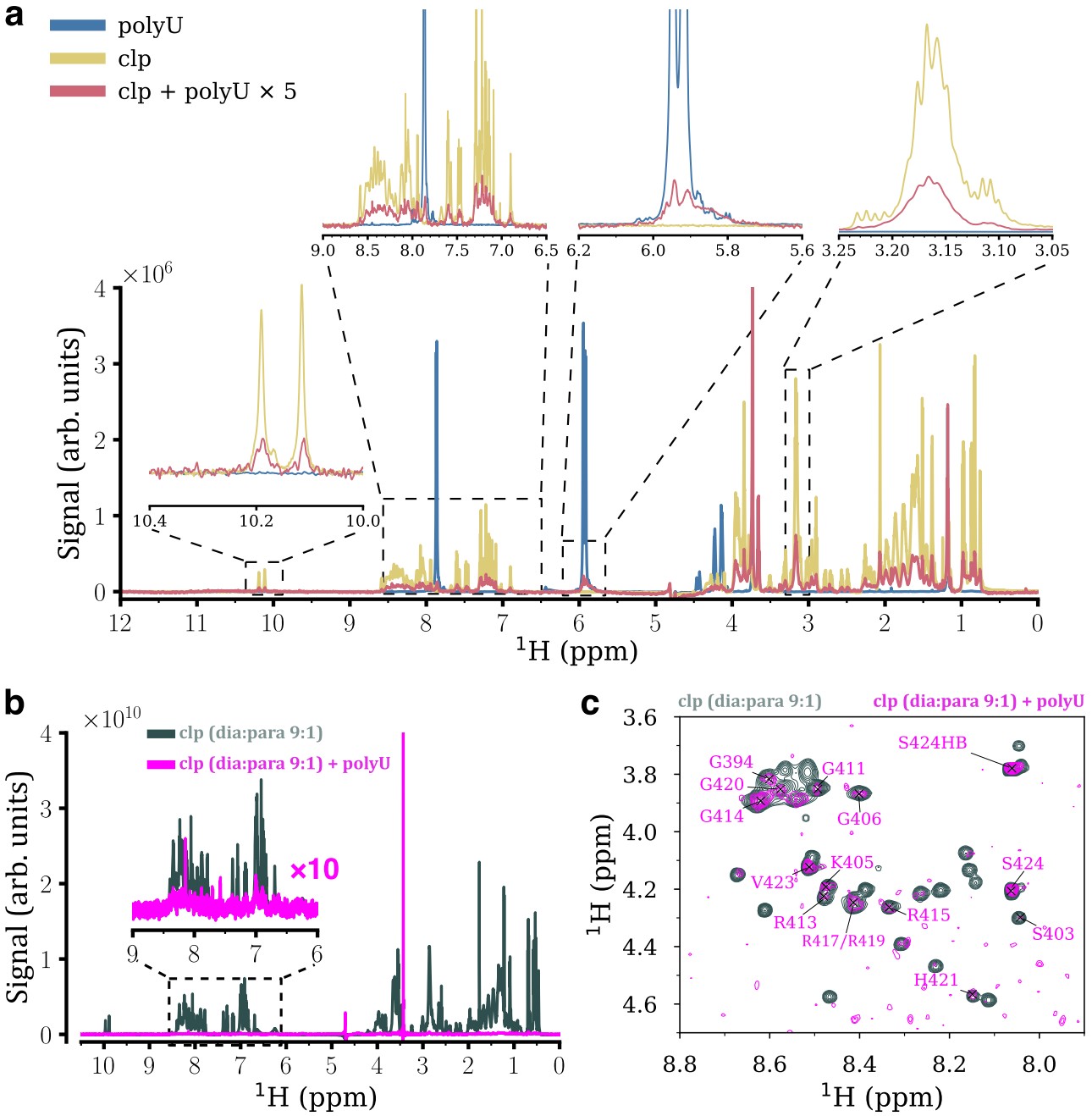

**Fig. 3 | NMR-based monitoring of the polyU RNA-induced LLPS of coilin peptide (clp).** **a** 1D ¹H NMR spectra of clp (gold), polyU RNA (blue) and after mixing both (red), showing reduction in signal intensities of clp and polyU at different regions of ¹H spectra, as a consequence of LLPS. For the sake of visibility, the ¹H spectrum of the mixture of clp and polyU is scaled up by a factor of 5. **b** 1D ¹H NMR of a mixture of dia- and paramagnetic clp, before (dark gray) and after (cyan) addition of polyU, showing a drastic reduction in signal intensity of clp caused by polyU-induced recruitment of dia- and paramagnetic clp into droplets and the resultant PRE effect. For the sake of visibility, the ¹H spectrum of the mixture of clp and polyU is scaled up by a factor of 10. Comparison with **a** shows a significant reduction in the intensity of polyU signals due to the PRE effect of paramagnetic clp, confirming the recruitment of clp to droplets. The signal at 3.43 ppm is due to a small molecule impurity of the used polyU batch. **c** 2D ¹H,¹H TOCSY spectra of a mixture of dia- and paramagnetic clp, before and after addition of polyU, showing residual peaks in the C-terminal and glycine regions. The TOCSY spectrum of the mixture of clp and polyU was obtained by five times higher number of NMR scans.

longer linker (MTSSL and PROXYL) and at two different labeling sites, the broadening caused by adding polyU is reproduced (Fig. 5b and Supplementary Fig. 9). Furthermore, for clp$_{PROXYL}$ the effect of dilution with the unlabeled clp has allowed to better distinguish the fast (*f*) and slow (*s*) components of the nitroxide spectrum upon addition of polyU (Fig. 5c). The broadening observed at room temperature is also observed in the Echo-Detected Field-Swept (ED-FS) spectra recorded at 34 GHz in frozen state (Supplementary Fig. 10). Together, the polyU-induced broadening of CW EPR signals in the three used spin probes

consistently indicate polyU-induced LLPS and its resultant restriction in the mobility of the labeled sites (with a maximum spectral width ranging from 3.2 mT to 6.1 mT).

Next, we employed Double-Electron Electron Resonance (DEER, known also as PELDOR) experiment[47] to monitor polyU-induced changes in intermolecular dipolar interactions in frozen solutions of singly labeled clp. The PELDOR method can provide information about intermolecular distances in the nanometer length scale (typically ranging between 1.5–8.0 nm)[48]. However, for the three labels used in this

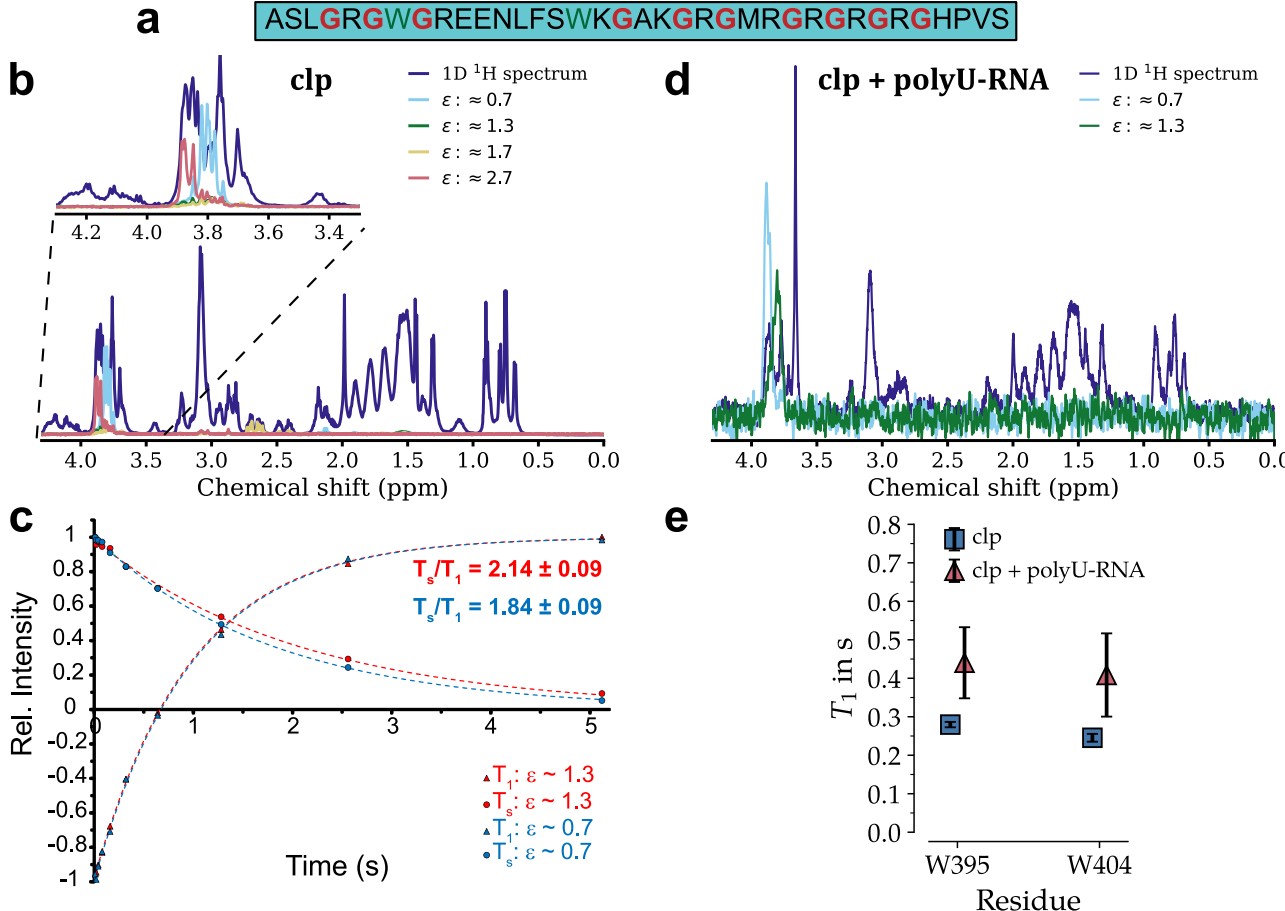

**Fig. 4 | Dynamics of coilin peptide (clp) probed via relaxation of glycine-based singlet state and tryptophan sidechains. a** Amino acid sequence of clp, highlighting its 10 glycine (red) and two tryptophan (dark green) residues. **b** Singlet-filtered 1D $^1$H spectra of clp enables selective detection of clp's glycines according to their HA-HA coupling regime ($\varepsilon = \Delta v/J$). The inset shows a close depiction of the spectrum at 3.35–4.25 ppm range, highlighting the singlet-filtered glycine signals. **c** Longitudinal spin-lattice ($T_1$) and singlet-state ($T_s$) relaxation curves of clp's glycines, shown for two groups of glycines with smallest $\varepsilon$ values. **d** Singlet-filtered 1D $^1$H spectra of clp after addition of polyU RNA allows accessing glycines even inside droplets. **e** $T_1$ relaxation times of W395 and W404 sidechain H$^\varepsilon$ resonances at 298 K. The fitted $T_1$ values for 0.5 mM clp (blue) and upon addition of 1 mg/mL polyU-RNA (red) are shown. Error bars indicate the standard fitting error obtained from the nonlinear least-squares minimization (see Supplementary Fig. 7). Source data are provided as a Source Data file.

**Table 1 | Singlet-state and spin-lattice relaxation times ($T_s$ and $T_1$) of different groups of glycine residues in coilin peptide (clp), measured at 298 and 278 K, before and after addition of polyU RNA**

|  | $T_s$ (s) 298 K (278 K) | $T_1$ (s) 298 K (278 K) | $T_s/T_1$ 298 K (278 K) | $T_s$ (s) 298 K | $T_1$ (s) 298 K | $T_s/T_1$ 298 K |
|---|---|---|---|---|---|---|
| Glycines | clp |  |  | clp + polyU |  |  |
| Group I: $\varepsilon = 0.7$ | 1.79 ± 0.03[a] (0.86 ± 0.05) | 0.98 ± 0.04 (1.37 ± 0.03) | 1.84 ± 0.08 (0.63 ± 0.04) | 1.84 ± 0.08 | 0.98 ± 0.04 | 1.87 ± 0.11 |
| Group II: $\varepsilon = 1.3$ | 2.08 ± 0.08 (0.97 ± 0.03) | 0.97 ± 0.02 (1.39 ± 0.02) | 2.14 ± 0.09 (0.70 ± 0.02) | 2.08 ± 0.10 | 1.05 ± 0.05 | 1.98 ± 0.13 |
| Group III: $\varepsilon = 1.7$ | 1.82 ± 0.23 (1.09 ± 0.15) | 0.93 ± 0.04 (1.40 ± 0.05) | 1.95 ± 0.26 (0.78 ± 0.11) | – | – | – |
| Group IV: $\varepsilon = 2.7$ | 2.23 ± 0.51 (1.24 ± 0.20) | 0.93 ± 0.05 (1.23 ± 0.05) | 2.40 ± 0.55 (1.01 ± 0.17) | – | – | – |

[a]The errors represent standard error of fitting.

study, the observed spin-pairs contribution to the dipolar interactions was remarkably weak (Supplementary Fig. 11 for MTSSL and PROXYL and Supplementary Fig. 12 for TOAC), precluding unambiguous description of intermolecular distance distributions. The very weak modulation depths observed in PELDOR experiments could be caused by the intrinsically disordered nature of coilin peptide and the consequent random distribution of intermolecular peptide orientations. Furthermore, the condensate phase induced upon addition of polyU is

likely to contain a significant fraction of clp molecules with intermolecular distances shorter than the distance detection limit of the PELDOR method (shorter than 1.5 nm).

For detecting hyperfine interacting 'electron-nuclei' between spin-labeled residue on clp and short-range distant nuclei (shorter than the range mentioned for the PELDOR/DEER experiments), we employed the HYSCORE method, by which the 'through-space' weak hyperfine electron–nucleus coupling could be detected. The unpaired

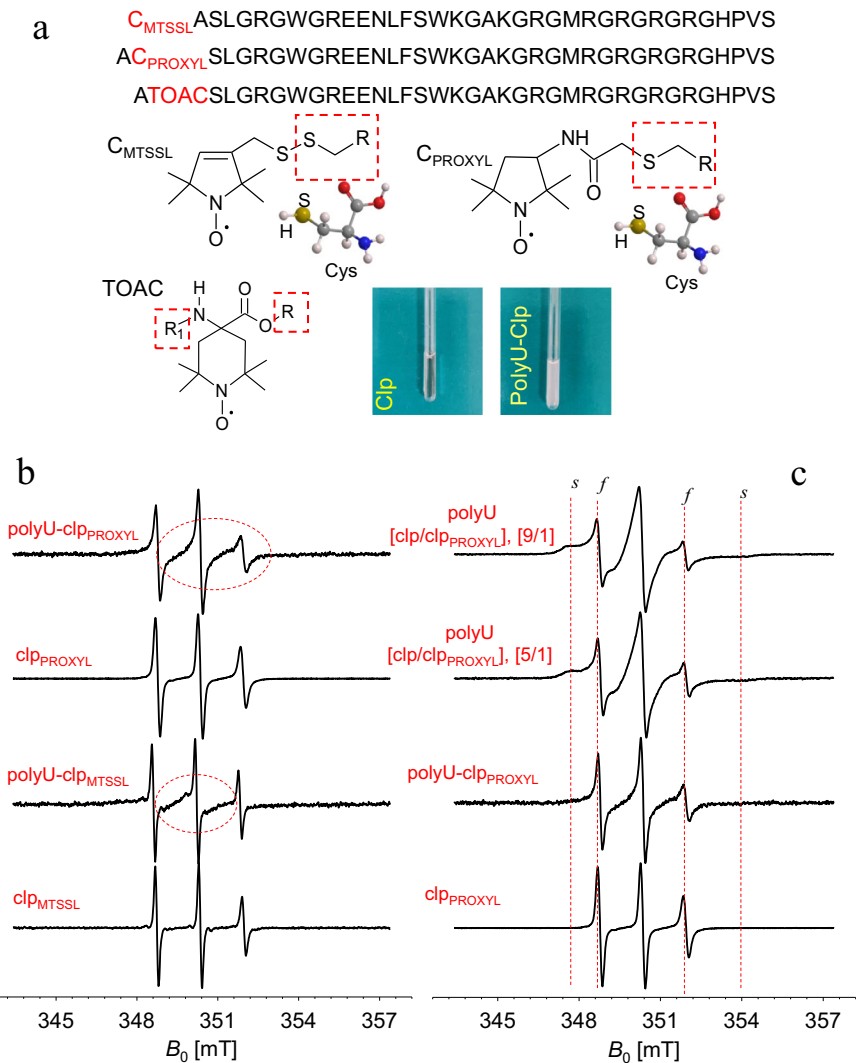

**Fig. 5 | Continuous-Wave (CW) EPR of the nitroxide-labeled coilin peptide before and after polyU RNA-induced LLPS. a** Sequence of the MTSSL-labeled coilin peptide, PROXYL-labeled peptide and TOAC-labeled peptide along with the chemical structure of MTSSL, PROXYL and TOAC moiety including its para-magnetic nitroxide group are shown. The pictures show the EPR X-band tubes containing clp and polyU/clp, respectively. **b** The CW EPR spectra of clp$_{PROXYL}$ and clp$_{MTSSL}$ show the line broadening upon addition of polyU. **c** the simultaneous formation of the slow (*s*) and fast (*f*) components of the nitroxide spectrum is even more pronounced upon dilution with unlabeled clp. The CW EPR spectra of clp$_{TOAC}$ are shown in Supplementary Fig. 8.

electron of the nitroxide probe (MTSSL, PROXYL or TOAC label) can exhibit strong hyperfine interactions (intra-molecular interactions) with $^{14}$N and $^{1}$H nuclei, but such strong interactions are out of the range for the HYSCORE experiments and therefore do not contribute to the measured signals. On the other hand, the inter-molecular weak inter-actions occurring between neighboring clp molecules may be detect-able if they are sufficiently close to each other. As shown in Fig. 6a, the HYSCORE spectrum of clp$_{MTSSL}$ in dilute phase, i.e. in the absence of LLPS induced by polyU, did not show any significant cross-peak typical of intermolecular electron-nucleus interaction in both quadrants (−,+) and (+,+). Interestingly, the HYSCORE spectrum of clp$_{MTSSL}$ in the polyU-induced condensed phase contained cross-peaks in the (+,+) quadrant (Fig. 6b)[49]. Such observation is validated by the 'surface projection' (inset for both experiments) where the frequencies observed upon LLPS are indicated by orange arrows. A similar trend is observed for clp$_{PROXYL}$ and polyU/clp$_{PROXYL}$ (Supplementary Fig. 13) and also for TOAC-labeled peptide (Supplementary Fig. 14).

## Discussion
Our data demonstrated the complex coacervation of the RG-rich coilin peptide (clp) together with polyU RNA leading to the formation of

relatively stable spherical droplets (Fig. 1). The use of fluorescently-labeled clp and nucleic acid-binding fluorescent dye (SYBR Gold) in fluorescence microscopy experiments and a mixture of diamagnetic and paramagnetic clp in NMR experiments confirmed the near com-plete recruitment of clp and polyU RNA peptides into the formed droplets (Figs. 1, 3 and Supplementary Fig. 1). The NMR relaxation of the glycine-based singlet-states of clp in dilute and polyU-induced condensed phases suggested that glycine residues retained a large level of their reorientational dynamics at hundreds of picoseconds timescale in the condensed phase (Fig. 4). On the other hand, the NMR $T_1$ relaxation of tryptophan sidechain protons suggested the partial rigidification of the two tryptophan residues of clp in the N-terminal region (Fig. 4 and Supplementary Fig. 7). The CW EPR data provided further support for the polyU-induced LLPS of clp and suggested the partial rigidification of the N-terminus of clp inside the condensed phase (Fig. 5). The pulsed EPR method HYSCORE showed that addition of polyU RNA decreased the intermolecular distances between spin-labeled clp molecules (Fig. 6).

Multiple types of interactions such as charge-charge, dipole-dipole, cation-π, π−π stacking, hydrogen bonding and hydrophobic interactions contribute to the network of multivalent interactions

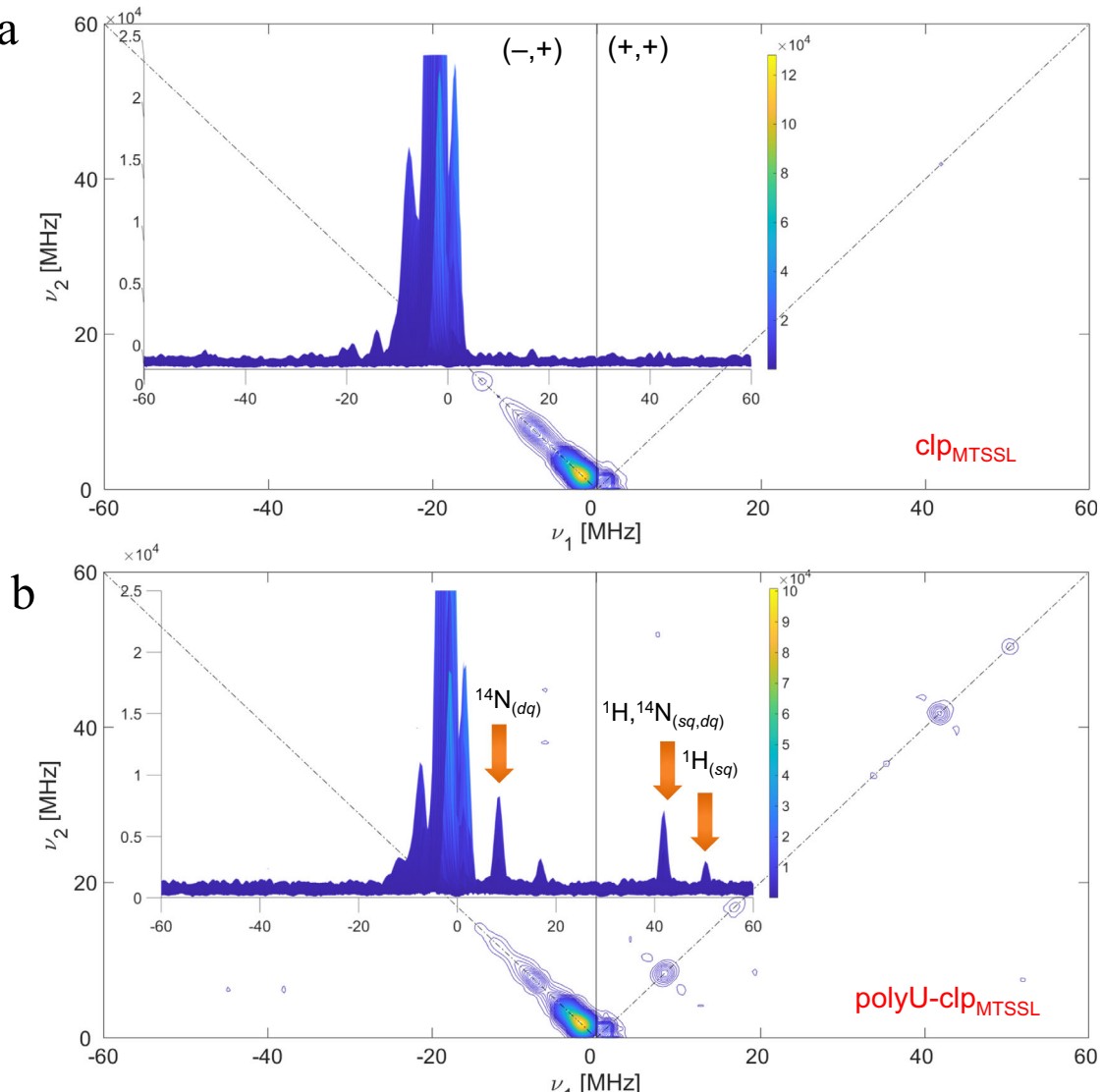

**Fig. 6 | Pulsed-EPR analysis of the MTSSL-labeled coilin peptide (clp$_{MTSSL}$) before and after polyU RNA-induced LLPS.** Hyperfine sublevel correlation (HYSCORE) spectroscopy: 2D plot for the HYSCORE experiments on clp (**a**) and clp/polyU (**b**). (−, +) and (+,+) quadrants show the diagonal and anti-diagonal peaks and cross-peaks generated by hyperfine interactions. For the (+,+) quadrant of the polyU/clp$_{MTSSL}$ sample, diagonal peaks can be assigned to DQ transition and combination frequencies of DQ-SQ of $^{14}$N-$^1$H. Such peaks are also shown with orange arrows on the 'surface projection' of the HYSCORE 2D plot.

involved in complex coacervation of ribonucleoproteins[50]. In case of the RG/RGG-rich unstructured regions ubiquitously found in RNA-interacting proteins[5–7], the positively charged arginine guanidinium groups are capable of long-range electrostatic interactions with the negatively charged phosphate groups of RNAs, as well as short-range directional cation-π and π-π interactions with the aromatic rings of purine and pyrimidine bases of RNAs[9,51]. While arginine residues play mechanistic roles in stabilizing RNA-protein interactions, the glycine residues seem to exploit their large conformational freedom and tune the internal fluidity of biomolecular condensates, as suggested previously[10,11]. Our data demonstrated that the RG/RGG-rich segment of coilin protein is intrinsically disordered and forms dynamic droplets after addition of polyU RNA. The glycine-based nuclear singlet-state approach employed in this study allowed probing dynamics of glycine residues in the polyU-induced condensed phase. Interestingly, despite the crowded environment within the formed droplets, as suggested by fluorescence intensity ratio between inside and outside droplets (Supplementary Fig. 1), several glycine residues exhibited almost the same singlet-state and longitudinal relaxation times $T_s$ and $T_1$ in the condensed phase as in the dilute solution. Importantly, the opposite temperature-dependence of $T_s$ and $T_1$ suggested that the motions of glycine residues occurred at intermediate timescales close to $T_1$ minimum where the sensitivity of $T_s/T_1$ ratio to dynamical changes is maximal[43–45]. Accordingly, the indistinguishable $T_s/T_1$ ratios in dilute solution and condensed phases strongly suggest that glycine residues of clp preserve their large local dynamics inside polyU-induced droplets. The dynamical behavior of glycines is in contrast with that of non-glycine residues, e.g. tryptophan sidechains, which exhibit considerable rigidification possibly due to π-π interactions with the aromatic pyrimidine rings of polyU. Our data are in general in line with previous NMR studies showing that IDPs remain disordered and mobile inside biomolecular condensates[18–21]. It furthermore highlights the specific role of remarkably mobile glycine residues as a molecular factor potentially supporting the internal fluidity and rapid component exchange of coilin-based droplets involved in the formation of RNA-splicing related CBs[35]. More generally, our data exemplifies the benefits of glycine-based singlet-state approach in monitoring dynamics of RG/RGG-rich domains within biomolecular condensates.

EPR spectroscopy has recently been used to detect the co-existence of dynamically distinct populations of a phase-separating

protein in its condensed phase[34]. Another recent study has demonstrated the promising applicability of a combined NMR and EPR approach to detect and quantify the LLPS of proteins and characterize their structure in both dispersed and condensed phases[33]. In addition, a combined NMR and EPR study has reported the effect of disordered RG/RGG regions on protein-RNA interactions[52]. In line with these recent studies, our CW-based line-shape analysis demonstrate the potential of EPR to monitor LLPS of proteins and provide complementary information with respect to the high-resolution NMR information. In particular, the CW EPR data with three different spin probes suggested the presence of a crowded phase inside RNA-induced clp droplets, in which rotational mobility of the labeled site in the N-terminus is considerably restricted. More importantly, our successful attempt to detect weak intermolecular interactions (electron-nucleus, e-N) in the condensed phase by using the HYSCORE technique, as shown in Fig. 6, provides a complementary approach to the more conventional dipolar electron–electron (e-e) coupling-based distance measurements in biomolecular condensates. Indeed, it paves the way to exploit this and other 2D-ESEEM techniques within the context of phase separation especially when the condensed phase is highly crowded, widening the range of putative hyperfine couplings with the paramagnetic probe. Notably, a more than 1000-fold increase in protein concentration inside droplets has been recently reported for an RGG-rich peptide undergoing RNA-induced phase separation[39], and high protein mass concentration inside droplets in the range of 100–700 mg/mL concentrations are reported for tau protein, FUS-LC, an RGG-rich peptide and ATXN3[39,53–55].

The coilin protein acts as the scaffold of CBs, a nuclear body supporting the RNA-splicing process[35]. The RG-rich coilin peptide studied here possesses the highest propensity for RNA-induced phase separation along the sequence of the full-length coilin (Fig. 1a) and is therefore expected to play a significant role in initiation of its RNA-induced phase separation and regulation of its droplet properties. Our data showing the large nanosecond dynamics of glycine residues inside coilin peptide droplets suggest a role for the RG-rich segment of coilin in supporting the high internal fluidity of CBs. Besides, the full-length coilin has a large content of lysine residues (~ 10 %), which could further support the internal fluidity of coilin droplets and CBs[10]. Notably, there is a Ser-rich region proximal to the RG-rich segment of full-length coilin, which can provide a Ser-phosphorylation-dependent regulatory mechanism for its phase separation through modulating intra- and intermolecular electrostatic interactions[56].

To summarize, we have studied the RNA-induced phase separation of an RG-rich fragment of coilin, the scaffold protein of Cajal body, through combining the complementary advantages of two high-resolution techniques, NMR and EPR. The CW and PELDOR EPR measurements demonstrated that the nitroxide spin-labeled coilin peptide forms dynamic droplets after addition of polyU RNA, inside which the coilin peptide is concentrated by two orders of magnitude. Despite such high concentration inside the formed droplets, however, the glycine-based singlet-state NMR relaxation data indicated that many glycine residues of coilin peptide almost completely preserved their large sub-nanoseconds dynamics inside the formed droplets. Our data suggest that the large dynamics of the RG-rich coilin peptide inside droplets may act as a potential molecular factor underlying the high dynamics and rapid component exchange of Cajal bodies, features crucially important for RNA splicing.

## Methods

### Sample preparation
The diamagnetic 36-residue long coilin peptide with the sequence H2N-ASLGRGWGREENLFSWKGAKGRGMRGRGRGRGHPVS-OH was purchased from Peptide Special Laboratory (PSL, Heidelberg, Germany) as well as the paramagnetic TOAC-labeled coilin peptide in which the TOAC label was inserted between residues Ala1 and Ser2.

The PROXYL-labeled coilin peptide in which the PROXYL label was attached to a cysteine residue inserted between residues Ala1 and Ser2 was purchased from the same company. For MTSSL-labeled coilin peptide, the coilin peptide with an additional cysteine residue at position 0 was synthesized on-site through solid-phase chemical synthesis (see Supplementary Data file) and the MTSSL labeling was achieved through a standard protocol. Briefly, 6 mg synthetic Cys0-coilin peptide was dissolved in 4 mL of 90% 200 mM HEPES buffer (pH = 7.60) and 10% dimethylformamide (DMF). 2.67 mL (1.0 µmol) of peptide solution was then mixed with 13.5 µmol MTSSL (Toronto Research Chemicals) dissolved in 634 µL DMF and incubated for 50 minutes at 20 °C in the dark, followed by RP-HPLC purification and lyophilization of the MTSSL-labeled peptide. Atto647 fluorescent labeling of coilin peptide was done by mixing 690 µL (211 nmol) of 306 µM Cys0-coilin peptide in 90% 200 mM HEPES buffer (pH = 7.60), 10% DMF with 422 nmol Atto647 maleimide (ATTO-TEC #AD 647-41, dissolved at 2.5 mM in DMF). This mixture was incubated for 50 min at 20 °C in the dark, followed by RP-HPLC purification and lyophilization of the fully labeled Atto647-Cys0-clp. The potassium salt of poly-uridylic acid (polyU RNA, MW: 600-1000 kDa, ~ 2000–3300 residues) was from Sigma-Aldrich. The nucleic acid-binding fluorescence dye SYBR Gold was from Invitrogen (10,000 times concentrated in DMSO). To induce phase separation of coilin peptide at room temperature, 10 µL of a 6 mg/mL polyU RNA solution was added to 50 µL of 0.48 mM coilin peptide solution, both in 50 mM sodium phosphate buffer (pH 6.5) containing 150 mM sodium chloride. The final concentrations were 1 mg/mL for polyU RNA and 0.4 mM for coilin peptide. Samples were prepared fresh for all our experiments (microscopy and NMR and EPR spectroscopy), as many times as it was needed. The reproducibility of phase separation was confirmed by inspection and microscopy and spectroscopy data. All the instances of repeated experiments were successful.

### DIC and fluorescence microscopy
Differential interference contrast (DIC) and fluorescence images were acquired using a Zeiss LSM780 confocal microscope with a C-Apochromat 40x/1.20 W Korr FCS M27 objective and processed using FIJI, the open source platform for biological image analysis[57]. For imaging, SYBR Gold (1:10000 dilution) was added to the phase separated samples containing a mixture of 1 mg/mL polyU RNA and 0.4 mM coilin peptide, as described above. The movement of selected droplets with time was followed using the TrackMate 7 Image J plugin[58].

For the colocalization experiments, phase separation was induced as described above but a 1:9 mixture of ATTO647-labeled coilin peptide and unlabeled coilin peptide was used. The masks were defined around objects with a diameter larger than 3 µm, using the Otsu threshold method implemented in FIJI[59]. Intensity ratios (IR) were calculated based on the mean intensity of SYBR Gold and ATTO647 fluorescence inside and outside of the condensates. Profiles of SYBR Gold and ATTO647 intensity across droplets were determined by performing line scans in FIJI.

### NMR experiments
NMR experiments were conducted on Bruker (Germany) spectrometers with proton Larmor frequencies of 600 and 800 MHz equipped with helium- or nitrogen-cooled (prodigy) cryoprobes. Unless specified otherwise, all NMR experiments were conducted at 298 K. The reference 2D $^1H,^1H$ TOCSY and NOESY and natural abundance 2D $^{15}N,^1H$ and $^{13}C,^1H$ HSQC spectra were acquired using samples containing 0.4 mM of coilin peptide, 50 mM sodium phosphate (pH 6.5) and 150 mM sodium chloride in water containing 5%, 10% (for singlet-filtered- or sf-TOCSY experiments) or 100% (for $^{13}C,^1H$ HSQC and singlet-filtered measurements) $D_2O$. In paramagnetic NMR experiments, a 9:1 mixture containing 9:1 (mass) concentration ratio of diamagnetic:paramagnetic (TOAC-labeled) coilin peptide was used. NMR measurements of the

phase separated coilin peptide were performed shortly (in 5-10 minutes) after addition of 1 mg/mL polyU RNA to the above-mentioned samples. The time-domain data in 2D $^1$H,$^1$H TOCSY and NOESY experiments contained 2048 and 800 or 600 complex data points, respectively in t$_2$ and t$_1$ dimensions. The corresponding values for the 2D $^{15}$N,$^1$H and $^{13}$C,$^1$H HSQC experiments were 1024 × 256 and 1024 × 512, respectively. Two mixing times of 60 and 75 ms for the TOCSY and 200 and 300 ms for the NOESY measurements were used. For TOCSY mixing, a DIPSI2 sequence was employed. The longitudinal spin-lattice ($T_1$) relaxation time of the nitrogen-attached proton of the indole ring of tryptophan residues were measured through saturation-recovery experiments, in which a 100 Hz continuous-wave (CW) irradiation applied on-resonance during recycle delays of 15–18 s fully saturated the signal, and after eight variable delays with durations of 0.04-5.12 s, the recovered signal intensity was measured through standard 1D $^1$H experiment with W5 WATERGATE elements. The fitting of recovered intensity vs delay time to a single-exponential recovery equation provided the $T_1$ relaxation time.

The glycine-based singlet-state measurements were conducted at 600 MHz proton Larmor frequency. Singlet-filtered 1D $^1$H and pseudo-2D longitudinal spin-lattice ($T_1$) and singlet-state ($T_s$) relaxation experiments were performed using the $gc$M2S sequence, as described previously[40]. Four sets of $n_1$, $n_2$, τ and Δ, suitable for glycine HA pairs with J/Δν ratios ranging from 0.7 to 2.7 (see Supplementary Table 1) were calculated and optimized. The sf-TOCSY spectra[40] were measured using the same TOCSY parameters as described above, a mixing time of 75 ms and singlet-state related parameters reported in Supplementary Table 1. In singlet-state-based $T_1$ and $T_s$ experiments, 10 relaxation delays exponentially sampling the range between 10 ms and 5.12 s were used. A total recycle delay of ca. 5 s was used in relaxation experiments.

Resonances of the clp peptide were assigned using the 2D $^1$H,$^1$H TOCSY and NOESY spectra to derive intra-residue and inter-residue correlations and to determine the amino acid residue type. Corresponding cross-peaks from the $^{13}$C,$^1$H HSQC spectrum were used to confirm this amino acid residue type assignment. Resonances in the $^{15}$N,$^1$H HSQC were assigned based on the $^1$H$^N$ assignment obtained from the homonuclear 2D spectra. POKY (Build 20220825) was used for the analysis described above[60]. Secondary structure propensities were obtained from the assigned chemical shifts using TALOS-N (Version 4.12 Rev 2015.147.15.40)[61].

### EPR experiments

CW EPR spectroscopy experiments were conducted on a Bruker EMX spectrometer operating at X-band frequencies (~9.48 GHz) using a standard rectangular Bruker EPR cavity (ER4102ST). All measurements were recorded at room temperature with 100-kHz field modulation, 0.2 G and 2.0 modulation amplitude.

Pulsed-EPR measurements were performed on a Bruker Elexsys E580 Q-band spectrometer or a Bruker spectrometer equipped with arbitrary waveform generator pulse channel. A temperature of 50 K was achieved by cryo-free Oxford system and a temperature controller. A sample tube loaded with 60 μL of sample was shock-frozen in liquid nitrogen and subsequently inserted into a Q-band resonator for 3-mm-outer-diameter sample tubes. An echo-detected field-swept EPR spectrum was acquired using a Hahn-echo sequence. Measurements on the Bruker EleXsys E580 spectrometer were acquired with a pulse delay τ$_1$ of 204 ns and a dead time delay of 100 ns. The pulse sequence for the four-pulse DEER experiment was π/2$_{obs}$ − τ$_1$ − π$_{obs}$ − t − π$_{pump}$ − (τ$_1$ + τ$_2$ − t$_1$) − π$_{ob}$ − τ$_2$. The pump pulse was applied on the spectral maximum and the observer pulses were applied at a frequency offset of 55 MHz. All traces were acquired using 10-ns pulses and eight-step phase cycling.

2D HYSCORE spectroscopy was performed at 50 K on a Bruker EleXsys E580 spectrometer. The measurements were conducted by applying a π/2 − τ − π/2 − t$_1$ − π − t$_2$ − π/2 − τ echo pulse sequence with

π/2 pulse length of 20 ns and an inversion π pulse length of 40 ns with time delays, τ, of 148, 172, and 204 ns to avoid blind-spot artifacts. The 2D HYSCORE time-domain data were recorded by measuring the echo amplitude as a function of dimensions t$_1$ and t$_2$, incremented in steps of 8 ns. The pulse sequence was repeated in an 8-step phase cycling procedure to avoid undesired echoes. The obtained spectra were then processed by (i) subtracting the background decay using a polynomial function, (ii) zero filling to 1024 points, (iii) tapering using a Hamming window, and by (iv) Fourier transforming the data in both t$_1$ and t$_2$ dimensions.

### Reporting summary

Further information on research design is available in the Nature Portfolio Reporting Summary linked to this article.

## Data availability

The sequence of protein coilin was taken from the UniProt entry with the code, P38432. Source data are provided with this paper.

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

## Acknowledgements

N.R.-G. acknowledges the Deutsche Forschungsgemeinschaft (German Research Foundation, DFG) for research grant RE 3655/2-3. For the EPR experiments the financial support from the IR INFRANALYTICS FR2054 is gratefully acknowledged. Drs. Peter Lenart from "Facility of Light Microscopy, Max Planck Institute for Multidisciplinary Sciences" and Joachim Maier are acknowledged for their helps in fluorescence microscopy imaging.

## Author contributions

Conceptualization, Writing—original draft: NRG; Project administration, Supervision: NRG, GS; Methodology, Investigation, Formal analysis, Data curation, Visualization: NRG, GS, DS, MK; Resources: NRG, GS, KO, RB, CG, DW; Writing-review & editing: NRG, GS, DS, RB, CG, DW, MK, KO.

## Funding

## Competing interests

The authors declare no competing interest.
