## [Peer Review File · Nature Communications]

REVIEWER COMMENTS

Reviewer #1 (Remarks to the Author):

Using NMR and EPR, authors have reported the dynamics of Gly and Trp side chains inside the LLPS formed by the RG-rich peptide region of coilin that consists of Cajal bodies. In particular, the recently developed glycine-based singlet-filtered approach demonstrated a high level of nanosecond-scale dynamics of Gly residues in the LLPS. Furthermore, the EPR experiments suggested that the peptides were concentrated approximately 1000-fold in the interior of the LLPS. However, the enrichment ratio, i.e., droplet-phase/homogeneous-phase, of the peptide is much higher than those of other proteins (e.g., 100-fold in ATXN3 LLPS and FUS-LC LLPS according to Raman imaging, *Chem Sci* 2021, 12, 7411; *JPC Lett* 2022, 13, 5692-5697). Additional verification through other approaches is necessary for the EPR-based estimation of enrichment ratio. Overall, it cannot be said that a sufficient understanding regarding the internal structure and dynamics of the peptide in the droplets has been achieved. I believe that this manuscript should be submitted to a specialized journal on biophysics.

Minor comments:

- 1) Abstract, line 28: nuclear magnetic resonance (NMR)
- 2) Page 7, line 132: Did 500 mM NaCl drive the protein toward phase separation? Based on the context, I gathered that it did not form LLPS.
- 3) Page 10, line 215–218: Same sentences have been mentioned twice.
- 4) Ref 4: Volume number and manuscript ID are missing.

Reviewer #2 (Remarks to the Author):

In this work, authors studied RNA-induced liquid-liquid phase separation of the arginine-glycine (RG)-rich segment of coilin (36 residue peptide), the major component of Cajal body (CB) using NMR and EPR spectroscopy as the major tools. Using fluorescence microscopy and NMR experiments they demonstrate near complete recruitment of polyU RNA and coilin peptides into droplets. Further NMR experiments suggested that that glycine residues retained a large level of their reorientational dynamics at hundreds of picoseconds timescale and a partial rigidification of the two tryptophan residues. Electron paramagnetic resonance (EPR) experiments suggested partial rigidification of the N-terminus and PELDOR and HYSCORE are interpreted as the addition of polyU RNA decreases the intermolecular distances between spin-labelled coilin peptides in the condensed phase. Overall, this is a very interesting study on an emerging and highly significant phenomenon of phase separation employing a range of techniques to visualize and characterize the several aspects including the formation of droplets and dynamics inside.

My comments are more centered on the EPR part of the study. Unfortunately, the data quality is way below the acceptable level and the results are heavily overinterpreted.

1. Figure 4B, right column: The CW EPR spectrum of the coilin peptide + polyU RNA sample has a very poor S/N. Also, the simulation from easyspin does not fit the data at all and the authors seriously discuss the changes in correlation times extracted from simulations. The authors need to make a better spectrum and redo the simulations.

2. Figure 5A, PELDOR data: Very poor S/N, very short time window and very small modulation depth make this data and the extracted distances useless. What was the labeling efficiency and spin concentration in the sample? Why could the authors not measure for longer time window (especially as there could be longer distances present)? Looking into the experimental setup, it is seen that they used 10 ns (rectangular?) pulses with a 55 MHz offset between pump and observer pulses, which might not be the optimal setup considering the overlap of the excitation bandwidths from the strong pulses used. In any case, the authors need to make better quality data (better S/N, modulation depth and longer time window).

How were the distances determined? The authors write (line 318, page 15) ‘...resulting in a distance distribution upon Fourier Transformation and Tikhonov regularization...’. At which step authors employed Fourier Transformation of the data (which is not required for Tikhonov regularization)? The distributions must be shown with an error bound.

Any interpretation of the data and the conclusions derived (average intermolecular distances and local concentration etc.,) are completely unreliable.

3. From the HYSORE data authors discuss intermolecular hyperfine interactions in the droplets. This suggests distances in Angstrom range between coilin peptides at the interacting interface. Would this agree with the average peptide concentration the authors estimated inside the droplet? A more detailed discussion is required to interpret the observed hyperfine couplings and interpeptide interaction.

4. Please discuss in more details the relevance of the outcome of this study in view of the full-length coilin and RNA splicing or other functions inside the cell.

Reviewer #3 (Remarks to the Author):

Sicoli et al. found for the first time a arginine-glycine rich segment of coilin, which is the main protein component of membrane less Cajal vesicles and which shows LLPS under in vitro conditions. The authors used state-of-the-art and advanced biophysical methods to characterize the condensed phase induced polyU addition. LLPS is currently of high biological relevance but very complex to study at the here presented molecular and dynamic level. The combination of NMR and EPR data generated additive value to understand the dynamic properties of the system. The authors further succeeded in using NMR single-state relaxation to a complex protein system to derive otherwise not accessible dynamic information, which is an exciting application of this method sofar more popular more for NMR scientists from physics and theory. This is a very important contribution to the field and should be considered for publication after addressing the following concerns.

1) I have one general concern: can the authors completely exclude contributions from the dilute phase after LLPS induced by polyU had been occurred. This is the core assumption for many observations attributed to the condensed phase e.g. lines 256-258. The authors presented various arguments but did

not mention for example that the ratio 1:9 TAOC labeled clp : clp was enough to almost completely broaden both the clp and polyU resonances by PRE. I also suggest a better depiction of the 1D spectra to see the missing resonances upon addition of polyU: same ppm range in Fig. 3D as in Fig. 3B and in the supplement separate close up depictions of the spectra from Fig. 2A for 0 ppm – 4.5 ppm and 5 ppm – 11 ppm of clp, polyU (so far missing), and the mixture and all 1D spectra at the same noise level. I expect that only few resonances (e.g. of glycine) remain visible because of their fortunate relaxation properties even in the droplets and all other resonances vanish.

Minor points:

2) line 135: Did the authors also try other ratios than 8:1 RNA/clp without observing LLPS? This would strengthen the point of matching compensating charges of polyU and clp to induce droplet formation. Please provide additionally at lines 133 – 141 or in M&M the volume of the 1 mg/mL polyU solution added to the 60 µL of 0.4 mM clp. Otherwise it is not comprehensible how a matching ratio of positive and negative charges could have been achieved from the polydisperse polyU sample with 2000 – 3300 nucleotides.

3) line 188 – 190: the PRE effect on both clp and polyU resonances is obvious from Fig. 2D and the conclusion is convincing but the depicted 1D of the mixture (magenta) might not be correctly scaled. The resonance at 3.4 ppm is not present in the spectra presented in Fig. 2A and is so not part of clp or polyU but was probably used for the y-scaling. At least the amide protons of those Gly and Arg residues remaining in the TOCSY of the mixture (Fig. 2E) should be visible in the 1D spectrum.

4) Table S1: what kind of experiment is gcM2S?

5) The quality of Fig. 4A can be improved, where the NO-label looks like copy/paste and in the droplet like insufficient removal of the transparent color.

6) typo ref. 33 in line 547: +-

We thank the three reviewers of our manuscript for their encouraging comments and constructive criticisms and suggestions, which in our view helped us to significantly improve our manuscript. Below, please find our detailed response to their comments.

REVIEWER COMMENTS

Reviewer #1 (Remarks to the Author):

Using NMR and EPR, authors have reported the dynamics of Gly and Trp side chains inside the LLPS formed by the RG-rich peptide region of coilin that consists of Cajal bodies. In particular, the recently developed glycine-based singlet-filtered approach demonstrated a high level of nanosecond-scale dynamics of Gly residues in the LLPS. Furthermore, the EPR experiments suggested that the peptides were concentrated approximately 1000-fold in the interior of the LLPS. However, the enrichment ratio, i.e., droplet-phase/homogeneous-phase, of the peptide is much higher than those of other proteins (e.g., 100-fold in ATXN3 LLPS and FUS-LC LLPS according to Raman imaging, Chem Sci 2021, 12, 7411; JPC Lett 2022, 13, 5692-5697). Additional verification through other approaches is necessary for the EPR-based estimation of enrichment ratio.

Reply: We thank the reviewer for raising this point and introducing interesting references. We would also like to draw the reviewer's attention to a more recent reference reporting around 1250-fold concentration ratio for an RGG-rich peptide (Yokosawa et al. Chem Phys Lett, 2023, 826: 140634). The 1000-fold concentration of the clp peptide inside droplets suggested in our study was based on our previous PELDOR-derived distance distribution assuming an average inter-spin distance of 1.6 nm and represented a semi-quantitative estimate of the clp concentration inside droplets. Now, following the critical comments made by the second reviewer, we have improved the quality of our PELDOR data by using the clp peptide labeled with different nitroxide spin probes. Based on our new PELDOR data (see below, and updated Fig. 5), we revise our estimated concentration ratio to 160, which is close to ratio around 140 or 150 reported in references mentioned by the reviewer and ref. 49 (Pantoja et al., Angew Chem, 2023) and smaller than the ratio of 1250 reported in Yokosawa et al. (see above). Interestingly, the mass concentration of clp inside droplets is estimated to be 250 mg/mL, similar to values reported in the four references mentioned above: 533-647 mg/ml (for ATXN3, depending on crowding agent), 373 mg/ml (for FUS-LC), 145 mg/ml (for tau protein) and 450 mg/ml (for RGG-rich peptide). We would however like to emphasize that the reported value represents only a semi-quantitative measure of concentration and should not be taken as a rigorous quantification of clp concentration inside droplets. To address the reviewer's comment, we have updated the concentration ratio in the text, added the following text to Results, pages 15-16, lines 330-335, and included the above mentioned references:

“A more than 1000-fold increase in protein concentration inside droplets has been recently reported for an RGG-rich peptide undergoing RNA-induced phase separation (ref. 39). The mass concentration of coilin peptide within droplets is 250 mg/mL, which is similar to the range of 100-700 mg/mL concentrations reported for tau protein, FUS-LC, an RGG-rich peptide and ATXN3 inside droplets (refs. 39, 49, 50, 51).”

Overall, it cannot be said that a sufficient understanding regarding the internal structure and dynamics of the peptide in the droplets has been achieved. I believe that this manuscript should be submitted to a specialized journal on biophysics.

Reply: With all due respect, we would like to highlight a few points of our disagreement with the reviewer: 1) Our study reports, to our knowledge, the first example of utilizing glycine-based singlet-state NMR in monitoring protein dynamics inside droplets. Considering the large content of glycine

residues in phase separating proteins, their relatively large mobility compared to other amino acid residues and their suggested role as spacers enhancing the fluidity of phase-separated droplets, we believe that the novel glycine-based singlet-state approach of our study is a powerful tool to access protein dynamics inside condensed phases. 2) Using two different EPR methods PELDOR and HYSCORE with complementary length-scale accessibility, we cover a broad range of intermolecular distances within droplets ranging from several Angstroms (accessed by HYSCORE) to 1.5-8.0 nanometers (accessed by PELDOR). To our knowledge, our study provides the first example of using HYSCORE in detecting and monitoring biomolecular phase separation. 3) The combined use of NMR and EPR demonstrates the “large dynamics of clp at glycine residues” (through glycine singlet-state approach) within the “highly crowded environment of clp droplets” (through EPR methods). We think that these points represent significant advancements in this rapidly growing field and justify the publication of this study in a journal with more general readership like Nature Communications.

Minor comments:

1) Abstract, line 28: nuclear magnetic resonance (NMR)

Reply: Done.

2) Page 7, line 132: Did 500 mM NaCl drive the protein toward phase separation? Based on the context, I gathered that it did not form LLPS.

Reply: No, it did not. The reviewer gathered it rightly. As said in the text, it was not sufficient to drive clp towards phase separation.

3) Page 10, line 215–218: Same sentences have been mentioned twice.

Reply: Thanks for spotting it. Corrected.

4) Ref 4: Volume number and manuscript ID are missing.

Reply: Corrected.

Reviewer #2 (Remarks to the Author):

In this work, authors studied RNA-induced liquid-liquid phase separation of the arginine-glycine (RG)-rich segment of coilin (36 residue peptide), the major component of Cajal body (CB) using NMR and EPR spectroscopy as the major tools. Using fluorescence microscopy and NMR experiments they demonstrate near complete recruitment of polyU RNA and coilin peptides into droplets. Further NMR experiments suggested that that glycine residues retained a large level of their reorientational dynamics at hundreds of picoseconds timescale and a partial rigidification of the two tryptophan residues. Electron paramagnetic resonance (EPR) experiments suggested partial rigidification of the N-terminus and PELDOR and HYSCORE are interpreted as the addition of polyU RNA decreases the intermolecular distances between spin-labelled coilin peptides in the condensed phase. Overall, this is a very interesting study on an emerging and highly significant phenomenon of phase separation employing a range of techniques to visualize and characterize the several aspects including the formation of droplets and dynamics inside.

My comments are more centered on the EPR part of the study.

Unfortunately, the data quality is way below the acceptable level and the results are heavily overinterpreted.

Figure 4B, right column: The CW EPR spectrum of the coilin peptide + polyU RNA sample has a very poor S/N.

Reply: a) We thank the reviewer for raising this point about the poor S/N of the CW EPR spectra. We have been aware about such major drawback and in the revised manuscript we focus on such aspect. For the TOAC coilin it can be underlined that the weaker signal has been obtained upon addition of polyU to TOAC-Coilin. When clp undergoes LLPS, coilin separates into a highly concentrated dense phase and a concomitant dilute phase, in which dynamic exchange can take place over short timescale. Such effect can generate nitroxide spin-exchange, and the very high broadening significantly affects the signal of EPR signal. Furthermore, reduction of the TOAC label produces an EPR-silent species. Thus, by using a nominal concentration of 0.4 mM and detecting lower than 40 μM of nitroxide signal, it can be deduced that the choice of the spin label (TOAC) might be supported by data recorded on coilin labeled with different nitroxide-type spin probe (for further discussion about TOAC, please see below appendix 1). In order to minimize drawbacks related to such spin probe, coilin peptide has been labeled with ‘five membered-rings’ nitroxide spin probes (MTSSL and PROXYL) and also the labelled site has been varied in order to monitor different sources generating weak nitroxide signals. Additional information on TOAC drawbacks compared with the updated new data are provided also in the supporting information. Experiments recorded with the novel samples exhibit an improved S/N ratios both for CW and pulsed EPR experiments. Please see pages 13-16, lines 283-354, and Figs. 5-6.

Also, the simulation from easyspin does not fit the data at all, and the authors seriously discuss the changes in correlation times extracted from simulations.

Reply: b) We agree with the reviewer that the low S/N of CW EPR spectrum does not allow a reliable quantitative analysis, therefore we have omitted the fitting of the CW EPR spectra from the main text and removed any serious quantitative discussion of correlation times. However, the polyU-induced alterations in CW EPR spectra observed by using MTSSL and PROXYL labels (now reported in the new figure 5), the fact that addition of polyU clearly generated typical *slow* and *fast* components of the nitroxide spectra, and observation of a broadening of the nitroxide spectrum even in the frozen samples, are all, albeit qualitatively, in line with changes in hyperfine values and rotational correlation times caused by polyU-induced LLPS. Our discussion of CW EPR data is now centered more on the qualitative interpretation of data. Please see pages 13-14, lines 283-307.

The authors need to make a better spectrum and redo the simulations.

Reply: c) As mentioned above at point **b**, CW fitting have been omitted; comparison among PROXYL, MTSSL and TOAC labels can exhaustively show the differences induced when coilin undergoes LLPS. Even if the initial CW spectra have been affected by low S/N ratios, the observed broadening induced by polyU is highly reproducible.

Figure 5A, PELDOR data: Very poor S/N, very short time window and very small modulation depth make this data and the extracted distances useless.

Reply: d) We again thank the reviewer for raising this point about the poor S/N of the DEER/PELDOR data and technical suggestions. The novel DEER/PELDOR data have been recorded with larger window (with a longer d_2 delay for the pulse sequence) and the results confirmed that a fraction of condensates exhibit inter-spin distances within the PELDOR/DEER detectable range (1.5 to 8.0 nm).

What was the labeling efficiency and spin concentration in the sample?

Reply: e) For the novel new label, the nominal concentration is corresponding to the ‘detected nitroxides’ (parameters for the echo-detected field-swept correspond to the concentration mentioned in the figure captions); this trend also confirms the reduction observed by using TOAC label.

Why could the authors not measure for longer time window (especially as there could be longer distances present)?

Reply: f) As specified at point **d**, the novel PELDOR experiments have been recorded with longer time window, having detected a refocused echo with such longer delay by using MTSSL and PROXYL labels.

Looking into the experimental setup, it is seen that they used 10 ns (rectangular?) pulses with a 55 MHz offset between pump and observer pulses, which might not be the optimal setup considering the overlap of the excitation bandwidths from the strong pulses used.

Reply: g) Considering the spectral width of the nitroxide radical species at Q-band frequency, the set up used into the submitted manuscript are not producing a significant overlapping of the pump/observe pulses; furthermore, experiments recorded with the setup of 10/20 ns or 20/40 ns do not differ from the initial choice of setup.

In any case, the authors need to make better quality data (better S/N, modulation depth and longer time window). How were the distances determined? The authors write (line 318, page 15) ‘...resulting in a distance distribution upon Fourier Transformation and Tikhonov regularization...’. At which step authors employed Fourier Transformation of the data (which is not required for Tikhonov regularization)?

Reply: h) We apologize if the text describing the DeerAnalysis procedure was not clear; before Tikhonov regularization procedure, the dipolar evolution function and the corresponding Fourier transformation are analyzed by ‘Approximate Pake Transformation’ (APT). The regularization step is then applied simultaneously to the dipolar evolution function and to the corresponding Pake pattern(s).

The distributions must be shown with an error bound.

Reply: i) For clarity, and to avoid crowded figures, in figure 6 the distance distributions for PROXYL labels are reported, while in Supplementary information the validation step of the DeerAnalysis is added, indicating error and ‘mean-distance’.

Any interpretation of the data and the conclusions derived (average intermolecular distances and local concentration etc.) are completely unreliable.

Reply: j) The new set of PELDOR/DEER experiments using MTSSL and PROXYL labeled coilin confirm the weak signal detected by applying these pulse sequences and, as specified at the point d), there are different sources for such weak signals; however, distance distributions upon addition of polyU are different in any case, with respect to the clp peptide without polyU. We agree with the reviewer that the weak S/N ratio does not allow a rigorous quantitative determination of intermolecular distances and local concentrations. However, we would like to demonstrate the potential value of this technique as a semi-quantitative tool to probe the changes of intermolecular distances occurring upon LLPS, as exemplified in our report.

From the HYSORE data authors discuss intermolecular hyperfine interactions in the droplets. This suggests distances in Angstrom range between coilin peptides at the interacting interface. Would this agree with the average peptide concentration the authors estimated inside the droplet? A more detailed discussion is required to interpret the observed hyperfine couplings and interpeptide interaction.

Reply: k) HYSCORE and PELDOR/DEER experiments are indeed complementary concerning the range of detectable distances; in the case of HYSCORE electron-nuclei distances are detected, while for the PELDOR/DEER experiments, electron-electron distances are detected. When clp undergoes LLPS, the clp condensates may indeed generate intermolecular interactions. Assuming that the clp-clp interactions may cover distances shorter than the clp-polyU interactions, the novel HYSCORE experiments, with higher S/N, confirm the electron-nuclear interactions. In particular interaction with ^{14}N and ^1H nuclei, as indicated in figure 6.

Please discuss in more details the relevance of the outcome of this study in view of the full-length coilin and RNA splicing or other functions inside the cell.

Reply: Thanks for the suggestion. We have now added a paragraph into Discussion, page 20, lines 431-443 where we shortly discuss on the relevance of our study in the context of full-length coilin and RNA splicing-related functions, as follows:

“The coilin protein acts as the scaffold of Cajal bodies, a nuclear body supporting the RNA splicing process (ref. 35). The RG-rich coilin peptide studied here possesses the highest propensity for RNA-induced phase separation along the sequence of the full-length coilin (Fig. 1A) and is therefore expected to play a significant role in initiation of its RNA-induced phase separation and regulation of its droplet properties. Our data showing the large nanosecond dynamics of glycine residues inside coilin peptide droplets suggest a role for the RG-rich segment of coilin in supporting the high internal fluidity of Cajal bodies. Besides, the full-length coilin has a large content of lysine residues (~ 10 %), which could further support the internal fluidity of coilin droplets and Cajal bodies (ref. 10). Notably, there is a Ser-rich region proximal to the RG-rich segment of full-length coilin, which can provide a Ser-phosphorylation-dependent regulatory mechanism for its phase separation through modulating intra- and intermolecular electrostatic interactions (ref. 56).”

Reviewer #3 (Remarks to the Author):

Sicoli et al. found for the first time a arginine-glycine rich segment of coilin, which is the main protein component of membrane less Cajal vesicles and which shows LLPS under in vitro conditions. The authors used state-of-the-art and advanced biophysical methods to characterize the condensed phase induced polyU addition. LLPS is currently of high biological relevance but very complex to study at the here presented molecular and dynamic level. The combination of NMR and EPR data generated additive value to understand the dynamic properties of the system. The authors further succeeded in using NMR single-state relaxation to a complex protein system to derive otherwise not accessible dynamic information, which is an exciting application of this method sofar more popular more for NMR scientists from physics and theory. This is a very important contribution to the field and should be considered for publication after addressing the following concerns.

Reply: Thanks for positive encouraging remarks.

I have one general concern: can the authors completely exclude contributions from the dilute phase after LLPS induced by polyU had been occurred. This is the core assumption for many observations attributed to the condensed phase e.g. lines 256-258. The authors presented various arguments but did not mention for example that the ratio 1:9 TAOC labeled clp : clp was enough to almost completely broaden both the clp and polyU resonances by PRE.

Reply: Thanks for pointing to this argument further supporting the core assumption for our data interpretation. We had briefly mentioned this argument and have now further highlighted it in Results, page 9, lines 187-195:

“The almost complete loss of clp and polyU signals after phase separation (Fig. 3B) indicates nearly complete recruitment of clp and polyU molecules to droplets, which brings the paramagnetic TOAC-labelled clp molecules close to diamagnetic clp and polyU molecules and consequently causes a paramagnetic relaxation enhancement (PRE)-induced broadening of their NMR signals. A similar behaviour has been recently reported for the RNA-induced phase separation of an RGG-rich peptide, for which Raman microscopy data demonstrated the full peptide recruitment into formed droplets (ref. 39)”.

We would also like to draw the reviewer’s attention to a recent paper on RNA-induced phase separation of an RGG-rich peptide (Yokosawa et al. Chem Phys Lett, 2023, 826: 140634), in which it is shown through Raman microscopy technique that the RGG-rich peptide is entirely recruited into RNA-induced droplets. We have now included this paper as ref. 39 in our revised manuscript.

I also suggest a better depiction of the 1D spectra to see the missing resonances upon addition of polyU: same ppm range in Fig. 3D as in Fig. 3B and in the supplement separate close up depictions of the spectra from Fig. 2A for 0 ppm – 4.5 ppm and 5 ppm – 11 ppm of clp, polyU (so far missing), and the mixture and all 1D spectra at the same noise level. I expect that only few resonances (e.g. of glycine) remain visible because of their fortunate relaxation properties even in the droplets and all other resonances vanish.

Reply: We have followed the reviewer’s suggestions: a) In Fig. 3, panels B and D have been replaced with spectra in the same ppm range, now at 0-4.25 ppm. In Panel B, an Inset shows a close-up depiction of the region between 3.35 and 4.25 ppm. Since the number of scans used for standard 1D and singlet-filtered spectra were different, the noise levels were not the same for different spectra, b) We have included a new supplementary Figure, Fig. S1, in which 1D spectra of clp, polyU (now included in the revised version) and clp+polyU mixture are shown separately at 0-4.5 ppm (panel A) and 4.5-11 ppm (panel B). For the sake of signal visibility in our spectra shown in Fig. S1, we did not use the same scale on Y-axis.

Minor points:

2) line 135: Did the authors also try other ratios than 8:1 RNA/clp without observing LLPS? This would strengthen the point of matching compensating charges of polyU and clp to induce droplet formation. Please provide additionally at lines 133 – 141 or in M&M the volume of the 1 mg/mL polyU solution added to the 60 microL of 0.4 mM clp. Otherwise it is not comprehensible how a matching ratio of positive and negative charges could have been achieved from the polydisperse polyU sample with 2000 – 3300 nucleotides.

Reply: We did not systematically explore RNA:clp ratios, instead picked a concentration ratio close to 1, which according to ref. 12 (Alshareedah et al., JACS, 2019), is expected to induce (almost) maximal phase separation. The reported concentrations of 1 mg/mL for polyU and 0.4 mM for clp were final concentrations after addition of 10 uL of 6 mg/mL polyU solution to 50 uL of 0.48 mM clp solution. We have added this information to SI, page S1.

3) line 188 – 190: the PRE effect on both clp and polyU resonances is obvious from Fig. 2D and the conclusion is convincing but the depicted 1D of the mixture (magenta) might not be correctly scaled. The resonance at 3.4 ppm is not present in the spectra presented in Fig. 2A and is so not part of clp or polyU but was probably used for the y-scaling. At least the amide protons of those Gly and

Arg residues remaining in the TOCSY of the mixture (Fig. 2E) should be visible in the 1D spectrum.

Reply: Thanks for mentioning this point. In general, we agree with the reviewer that the 1D spectra shown in Fig. 2 are not clearly visible, therefore we decided to re-structure Fig. 2 and split it into two Figures 2 and 3. In the new Fig. 3, we have made the following changes: a) In panel A, we show the 1D ^1H spectra of three samples (clp alone, polyU alone and clp+polyU) with close-up depiction of some regions highlighting clp or polyU signal attenuation after polyU-induced phase separation. For the sake of improving signal visibility, we have scaled up the 1D spectrum of clp+polyU sample by a factor of 5. B) In panel B, we show the 1D ^1H spectra of clp (dia:para 9:1) and clp+polyU samples, with close-up depiction at the amide region where some residual signals (e.g. from Gly and Arg residues) could be seen. There, we have scaled up the 1D spectrum of clp+polyU sample by a factor of 10 to make the residual signals visible. Please note that the 2D TOCSY spectra shown in the old panel Fig. 2E (the new panel Fig. 3C) were acquired using 8 and 40 number of scans, respectively, for the clp and clp+polyU samples, and the peak signal intensities in the TOCSY spectrum of clp+polyU sample were very low compared to those of the clp sample. This is consistent with the very low intensity of signals in 1D ^1H spectrum of clp+polyU sample in new Fig. 3B. We have added this clarifying information to caption to Fig. 3.

In regard with the signal at 3.4 ppm: the NMR data for a mixture of dia and paramagnetic clp and polyU (shown in Fig. 3, new panels B and C) were obtained using a second batch of polyU, which showed an additional NMR signal at ca. 3.43 ppm (see Fig. L1 below). This signal is likely to be originated from a small molecule impurity of that batch. After mixing with clp (dia:para 9:1), the polyU signals were almost entirely lost, while the signal at 3.43 ppm survives. We have added this clarification point to caption to Fig. 3.

Figure L1. 1D ^1H spectra of polyU, before (blue) and after (red) addition of a mixture of dia- and paramagnetic clp. The signal at 3.43 ppm is already present in the spectrum of polyU sample (blue spectrum) and probably originated from a small-molecule impurity present in the polyU batch used for this experiment.

4) Table S1: what kind of experiment is gcM2S?

Reply: The gcM2S acronym stands for “general coupling Magnetization-to-Singlet”. This is a recently developed pulse sequence, cited as ref. 40 in the main text and ref. 2 in SI, for accessing singlet states in spin pairs coupled under different coupling regimes. We have added the following description to the footnote of Table S1:

“*. The gcM2S acronym stands for “general coupling Magnetization-to-Singlet”, referring to a recently developed pulse sequence cited above. This pulse sequence allows accessing singlet-states in spin-pair systems coupled over a broad range of coupling regimes.”

5) The quality of Fig. 4A can be improved, where the NO-label looks like copy/paste and in the droplet like insufficient removal of the transparent color.

Reply: Figure 4A (now as 5A) has been modified according to the reviewer suggestions.

6) typo ref. 33 in line 547: +-

Reply: Corrected.

Appendix 1:

A more extensive overview on TOAC label.

TOAC spin probe has been the first choice for labeling Coilin (clp) (*Scheme S1*). The advantage of TOAC over side chain-attached spin labels resides in the fact that the former is linked via a peptide bond; moreover, due to its cyclic structure, the molecule’s restricted mobility hampers rotation about side chain bonds. In contrast, side chain-attached spin labels (i.e. MTSSL, PROXYL) lead to higher conformational freedom, rendering the analysis of backbone conformation more difficult. (Schereier, S., Bozelli, J.-C., Marin, N., Vieira, R. F. F., Nakaie, C. “The spin label amino acid TOAC and its uses in studies of peptides: chemical, physicochemical, spectroscopic and conformational aspects” *Biophysical Reviews* **2012**, 4, 45-66).

Scheme S1

However, incorporation of TOAC via peptide coupling does not occur in good yield; often requiring multiple rounds of peptide coupling, due to the low nucleophilicity of the sterically hindered amino group. TOAC has a very limited range of backbone dihedral angles, creating a significant distortion of the secondary structure of proteins. Thus, even if this label is

exceptionally rigid and allows flexibility of the paramagnetic center only by flipping of the piperidine moiety, this rigidity also bears the potential of perturbing peptide secondary structures (**a**) Fielding, A. J., Concilio, M. G., Heaven, G., Hollas, M. A. "New developments in spin labels for pulsed dipolar EPR" *Molecules* 2014, 19, 19998-17025; **b**) Roser, P., Schmidt, M. J., Drescher, M., Summerer, D. "Site-directed spin labeling of proteins for distance measurements in vitro and in cells" *Organic and Biomolecular Chemistry* 2016, 14, 5468-5476). Furthermore, a critical step for the use of TOAC is the strict control of pH, in order to avoid the free radical to disproportionate to an oxoammonium cation and TEMPO-OH, both EPR silent molecules (*Scheme S2*) (Karim, C. B., Zhang, Z., Thomas, D. D. "Synthesis of TOAC spin-labeled proteins and reconstitution in lipid membranes" *Nature Protocols* 2007, 2, 42-49).

Scheme S2

The chemical and structural properties mentioned above may significantly affect the nominal concentration of TOAC-Coilin (Clp_{TOAC}) and reduce the efficiency in monitoring LLPS, upon the addition of polyU to Clp_{TOAC} . Additionally, upon the formation of 'droplets' (thus upon LLPS), the 'condensed phase' may bring in close proximity (below 0.1 nm) the nitroxide spin probes and the 'exchange coupling' can affect spectral features by strongly perturbing hyperfine and dipolar couplings.

In structurally simpler molecules (i.e. dinitroxide molecules) strong $J \gg A_N$ (exchange coupling \gg hyperfine coupling) between electron spins occur via through-bond pathways; these molecules exhibit five-lines EPR spectra; however, in many diradicals and in the scenario described in the LLPS of Clp_{TOAC} /polyU, a dynamic interconversion between conformations with weak exchange and strong exchange may occur; in such a case the spectral features can be closer to the mono-radical; also in this case, the dipolar effect (for PELDOR/DEER analysis) is drastically reduced (Eaton, S. S., Woodcock, L. B., Eaton, G. R. "Continuous wave electron paramagnetic resonance of nitroxide biradicals in fluid solution" *Concepts in Magnetic Resonance Part A* 2019, 47A, e21426).

The more crowded scenario upon LLPS can indeed be generated also by "multi-spins contribution" (known also as "spin-bath contribution"), derived by the formation of a condensed region.

REVIEWER COMMENTS

Reviewer #1 (Remarks to the Author):

The authors carefully responded to each reviewer's comment. In the end, I evaluate that this study using EPR and NMR has a high degree of novelty and provides important contribution to the field.

Minor comments:

1) Initially, I had doubts about estimating the concentration of coilin peptide in the droplet, and suggested that other methods be considered. Other reviewers also pointed out the poor signal-to-noise ratio of the PELDOR spectrum, that is used for measuring the PELDOR-based distance between spin labels. In the revised manuscript, the authors re-measured the PELDOR and revised the predicted clp concentration within the droplet from 1000-fold the initial concentration to 160-fold. Spectrum measurement with a high S/N is also required since the calculated value changes by one order of magnitude depending on the measurement accuracy. Regarding estimation and validation of intermolecular distance from the PELDOR, Fig. s11 is cited, but details such as the formula used are unknown. I think that distance calculation is based on dipole-dipole interaction between unpaired electrons of the spin probes, but some explanation is required in the method section (or supporting materials).

2) Abstract, line 35: clp -> coilin peptide

3) In Fig S4(A), it is necessary to check whether the direction of the arrow regarding S424CB is correct. No signal can be seen in the direction of the indicator stick.

Reviewer #2 (Remarks to the Author):

1. The authors have faithfully responded to the questions on continuous wave ESR spectroscopy experiments. They performed additional experiments with improved S/N and analysis to show that LLPS leads to restricted mobility of the spin labelled sites inside the droplets.

2. Unfortunately, the DEER/PELDOR part still remains as a major issue. The authors may refer to the guidelines given in the white paper for PELDOR data acquisition, analysis, presentation and interpretation.

Benchmark Test and Guidelines for DEER/PELDOR Experiments on Nitroxide-Labeled Biomolecules. J. Am. Chem. Soc. 2021, 143, 43, 17875–17890, <https://pubs.acs.org/doi/10.1021/jacs.1c07371>

3. The authors used the PELDOR data mostly used to support the clp peptide concentration inside the droplets. It would be sufficient if the authors can substantiate this conclusion with another approach. The PELDOR data as they are reported now are unreliable/unpublishable for the following reasons.

4. With just 1-2% modulation depth of the PELDOR data for nitroxide labeled samples, especially at the current S/N level, no reliable distances determination can be performed. This will be apparent if the authors follow the guidelines for data analysis in the above manuscript. Also, all the primary data are

missing and the distance distribution cannot be presented without an error bound in the manuscript (Figure 6c).

5. The overall errors in the experiments are reflected from the huge variation of the peptide concentration inside the droplets between the original submission and the revision (1000-fold vs. 160-fold!). The distances are also drastically different (due to poor data quality in both cases), which eventually lead to this discrepancy (as the authors used those distances to calculate the concentration). Unfortunately, the authors make serious claims based on unreliable PELDOR data sets, which is incorrect. The local concentration may be extracted from the decay of the background function of the PELDOR data (which the authors haven't attempted); however, it may still pose the challenge that signals coming from very short distances are selectively lost due to faster relaxation. From the NMR data the authors concluded that clp peptide is predominantly an intrinsically disordered peptide, with weak secondary structure propensities. It's likely that the random intermolecular peptide orientations make it difficult to separate the wanted inter-peptide distances (form factors) from the background contribution in line with the extremely small modulation depth values the authors observed.

Reviewer #3 (Remarks to the Author):

All concerns raised by this reviewer have been very well addressed in the revised manuscript in the main text and the modified figures. No further improvements are required.

We thank again the three reviewers for their constructive suggestions during evaluation of our manuscript. Below, please find our response to their latest comments.

REVIEWER COMMENTS

Reviewer #1 (Remarks to the Author):

The authors carefully responded to each reviewer's comment. In the end, I evaluate that this study using EPR and NMR has a high degree of novelty and provides important contribution to the field.

Reply: We sincerely appreciate the positive evaluation of our work by Reviewer 1.

Minor comments:

1) Initially, I had doubts about estimating the concentration of coilin peptide in the droplet, and suggested that other methods be considered. Other reviewers also pointed out the poor signal-to-noise ratio of the PELDOR spectrum, that is used for measuring the PELDOR-based distance between spin labels. In the revised manuscript, the authors re-measured the PELDOR and revised the predicted clp concentration within the droplet from 1000-fold the initial concentration to 160-fold. Spectrum measurement with a high S/N is also required since the calculated value changes by one order of magnitude depending on the measurement accuracy. Regarding estimation and validation of intermolecular distance from the PELDOR, Fig. s11 is cited, but details such as the formula used are unknown. I think that distance calculation is based on dipole-dipole interaction between unpaired electrons of the spin probes, but some explanation is required in the method section (or supporting materials).

Reply: We have now followed the Reviewer's suggestion and utilized fluorescence microscopy to evaluate clp peptide concentration ratio between condensed and dilute phases. Using a 9:1 mixture of unlabeled clp and clp labeled with a fluorescent tag (Atto647), we monitored the polyU-induced LLPS of clp and provided an estimate for the clp concentration ratio between inside and outside of the formed droplets (the intensity ratio of 204 ± 49 , taken as an estimate of concentration ratio). These data have now been included in supplementary Fig. S1 and in Results, page 7, lines 144-150, as:

“In addition, the combined use of fluorescently-labelled clp (with Atto647) and polyU-binding fluorescent dye confirmed the co-localization of clp and polyU in the formed droplets (Supplementary Fig. S1A). The calculated fluorescence intensity ratio between inside and outside of droplets was 204 ± 49 for Atto647-labeled clp and 55 ± 18 for SYBR Gold-bound polyU (Supplementary Fig. S1B,C), indicating that clp and polyU were almost entirely recruited to the formed droplets.”

However, due to technical limitations (e.g. the unknown level of collisional quenching occurring inside the condensed phase), these data do not allow us to reliably estimate the absolute concentration of coilin peptide inside droplets.

We also fully agree with the Reviewer (and Reviewer 2) about the effect of low S/N ratio on the accuracy of our PELDOR-based calculation of distance distributions. In this regard, we would like to underline that from several experiments with different spin probes recorded both at X- and Q-band frequencies where set up have been varied and optimized, the exponential dipole-dipole contribution, following the background correction, was substantially lower than 15 %, making it extremely challenging to quantitatively extract distance distributions. As pointed out by the Reviewer 2, the intrinsically disordered nature of coilin peptide, hence randomness of intermolecular peptide orientations, is probably one of the reasons to “*make it difficult to separate the wanted inter-peptide distances (from factors) from the background contribution*” and cause modulation depths being too

small for a reliable distance calculation. Another general restriction in PELDOR-based quantitative determination of concentration in condensed phases is the limited length scale sensitivity of PELDOR technique. For distances shorter than 15 angstroms, it is difficult to disentangle the remaining dipole-dipole contribution of signal modulation upon removing the exponential decay. For those reasons, our intention was to present our PELDOR-based distance distributions, not as a rigorous quantification of concentration, but only as a semi-quantitative measure enabling comparison with some recent reports of intra-droplet concentrations determined through single-droplet Raman microscopy method. Now, to address the valid concerns raised by both the Reviewers 1 and 2, we have removed all the reported distance distributions from the text and Figure 6, moved the raw PELDOR/DEER data into supplementary information (now as Figs. S11 and S12) and also modified the main text, page 15, lines 317-326, accordingly:

“However, for the three labels used in this study, the observed spin-pairs contribution to the dipolar interactions was remarkably weak (Supplementary Fig. S11 for MTSSL and PROXYL and Fig. S12 for TOAC), precluding unambiguous description of intermolecular distance distributions. The very weak modulation depths observed in PELDOR experiments could be caused by the intrinsically disordered nature of coilin peptide and the consequent random distribution of intermolecular peptide orientations. Furthermore, the condensate phase induced upon addition of polyU is likely to contain a significant fraction of clp molecules with intermolecular distances shorter than the distance detection limit of the PELDOR method (shorter than 1.5 nm).”

2) Abstract, line 35: clp -> coilin peptide

Reply: Done.

3) In Fig S4(A), it is necessary to check whether the direction of the arrow regarding S424CB is correct. No signal can be seen in the direction of the indicator stick.

Reply: Thanks for spotting it. The S424CB correlation peak had low intensity and was not visible at the contour level chosen to show all the CA correlation peaks. We have now put a “+” sign at the place of the S424CB peak and added an explanation to Figure caption (now it is Fig. S5A).

Reviewer #2 (Remarks to the Author):

1. The authors have faithfully responded to the questions on continuous wave ESR spectroscopy experiments. They performed additional experiments with improved S/N and analysis to show that LLPS leads to restricted mobility of the spin labelled sites inside the droplets.

Reply: Thanks for the positive evaluation by the Reviewer.

2. Unfortunately, the DEER/PELDOR part still remains as a major issue. The authors may refer to the guidelines given in the white paper for PELDOR data acquisition, analysis, presentation and interpretation.

Benchmark Test and Guidelines for DEER/PELDOR Experiments on Nitroxide-Labeled Biomolecules. *J. Am. Chem. Soc.* 2021, 143, 43, 17875–17890, <https://pubs.acs.org/doi/10.1021/jacs.1c07371>

Reply: Thanks for introducing the paper, which was indeed already included as reference 48 in our manuscript and partially used as a guideline in our analysis and semi-quantitative interpretation of

PELDOR/DEER data. Please see below how we have addressed the valid concerns raised by the Reviewer about our DEER/PELODR data.

3. The authors used the PELDOR data mostly used to support the clp peptide concentration inside the droplets. It would be sufficient if the authors can substantiate this conclusion with another approach. The PELDOR data as they are reported now are unreliable/unpublishable for the following reasons.

Reply: We have now followed the Reviewer's suggestion and utilized fluorescence microscopy to evaluate clp peptide concentration ratio between condensed and dilute phases. Using a 9:1 mixture of unlabeled clp and clp labeled with a fluorescent tag (Atto647), we monitored the polyU-induced LLPS of clp and provided an estimate for the clp concentration ratio between inside and outside of the formed droplets (the intensity ratio of 204 ± 49 , taken as an estimate of concentration ratio). These data have now been included in supplementary Fig. S1 and in Results, page 7, lines 144-150, as:

“In addition, the combined use of fluorescently-labelled clp (with Atto647) and polyU-binding fluorescent dye confirmed the co-localization of clp and polyU in the formed droplets (Supplementary Fig. S1A). The calculated fluorescence intensity ratio between inside and outside of droplets was 204 ± 49 for Atto647-labeled clp and 55 ± 18 for SYBR Gold-bound polyU (Supplementary Fig. S1B,C), indicating that clp and polyU were almost entirely recruited to the formed droplets.”

However, due to technical limitations (e.g. the unknown level of collisional quenching occurring inside the condensed phase), these data do not allow us to reliably estimate the absolute concentration of coilin peptide inside droplets.

4. With just 1-2% modulation depth of the PELDOR data for nitroxide labeled samples, especially at the current S/N level, no reliable distances determination can be performed. This will be apparent if the authors follow the guidelines for data analysis in the above manuscript. Also, all the primary data are missing and the distance distribution cannot be presented without an error bound in the manuscript (Figure 6c).

Reply: Please see below for our response.

5. The overall errors in the experiments are reflected from the huge variation of the peptide concentration inside the droplets between the original submission and the revision (1000-fold vs. 160-fold!). The distances are also drastically different (due to poor data quality in both cases), which eventually lead to this discrepancy (as the authors used those distances to calculate the concentration). Unfortunately, the authors make serious claims based on unreliable PELDOR data sets, which is incorrect. The local concentration may be extracted from the decay of the background function of the PELDOR data (which the authors haven't attempted); however, it may still pose the challenge that signals coming from very short distances are selectively lost due to faster relaxation. From the NMR data the authors concluded that clp peptide is predominantly an intrinsically disordered peptide, with weak secondary structure propensities. It's likely that the random intermolecular peptide orientations make it difficult to separate the wanted inter-peptide distances (form factors) from the background contribution in line with the extremely small modulation depth values the authors observed.

Reply to 4 and 5: We fully agree with the Reviewer about the effect of low S/N ratio and weak modulation depth on the accuracy of our PELDOR-based calculation of distance distributions. In this regard, we would like to underline that from several experiments with different spin probes recorded both at X- and Q-band frequencies where set up have been varied and optimized, the exponential dipole-dipole contribution, following the background correction, was substantially lower than 15 %, making it extremely challenging to quantitatively extract distance distributions. We also fully agree

with the nicely articulated arguments of the Reviewer regarding the issues caused by the intrinsically disordered nature of coilin peptide and the limited access of PELDOR technique to short distances. For the same reasons, our intention was to present our PELDOR-based distance distributions, not as a rigorous quantification of concentration, but only as a semi-quantitative measure enabling comparison with some recent reports of intra-droplet concentrations determined through single-droplet Raman microscopy method. Now, to address the valid concerns raised by the Reviewer, we have toned down our PELDOR-based statements by entirely removing all the reported distance distributions from the text and Figure 6, moving the raw PELDOR/DEER data into supplementary information (now as Figs. S11 and S12) and also modifying the main text, page 15, lines 317-326, accordingly:

“However, for the three labels used in this study, the observed spin-pairs contribution to the dipolar interactions was remarkably weak (Supplementary Fig. S11 for MTSSL and PROXYL and Fig. S12 for TOAC), precluding unambiguous description of intermolecular distance distributions. The very weak modulation depths observed in PELDOR experiments could be caused by the intrinsically disordered nature of coilin peptide and the consequent random distribution of intermolecular peptide orientations. Furthermore, the condensate phase induced upon addition of polyU is likely to contain a significant fraction of clp molecules with intermolecular distances shorter than the distance detection limit of the PELDOR method (shorter than 1.5 nm).”

Reviewer #3 (Remarks to the Author):

All concerns raised by this reviewer have been very well addressed in the revised manuscript in the main text and the modified figures. No further improvements are required.

Reply: We greatly appreciate the positive evaluation of our work by Reviewer 3.